# Low-nuclearity CuZn ensembles on ZnZrO$_x$ catalyze methanol synthesis from CO$_2$

Thaylan Pinheiro Araújo[1], Georgios Giannakakis [1], Jordi Morales-Vidal [2], Mikhail Agrachev[3], Zaira Ruiz-Bernal [4], Phil Preikschas [1], Tangsheng Zou[1], Frank Krumeich [5], Patrik O. Willi[1], Wendelin J. Stark[1], Robert N. Grass [1], Gunnar Jeschke[3], Sharon Mitchell [1], Núria López [2] & Javier Pérez-Ramírez [1] ✉

Metal promotion could unlock high performance in zinc-zirconium catalysts, ZnZrO$_x$, for CO$_2$ hydrogenation to methanol. Still, with most efforts devoted to costly palladium, the optimal metal choice and necessary atomic-level architecture remain unclear. Herein, we investigate the promotion of ZnZrO$_x$ catalysts with small amounts (0.5 mol%) of diverse hydrogenation metals (Re, Co, Au, Ni, Rh, Ag, Ir, Ru, Pt, Pd, and Cu) prepared via a standardized flame spray pyrolysis approach. Cu emerges as the most effective promoter, doubling methanol productivity. Operando X-ray absorption, infrared, and electron paramagnetic resonance spectroscopic analyses and density functional theory simulations reveal that Cu$^0$ species form Zn-rich low-nuclearity CuZn clusters on the ZrO$_2$ surface during reaction, which correlates with the generation of oxygen vacancies in their vicinity. Mechanistic studies demonstrate that this catalytic ensemble promotes the rapid hydrogenation of intermediate formate into methanol while effectively suppressing CO production, showcasing the potential of low-nuclearity metal ensembles in CO$_2$-based methanol synthesis.

Mixed zinc-zirconium oxides, ZnZrO$_x$, have emerged as cost-effective and earth-abundant catalysts for CO$_2$ hydrogenation to methanol (CO$_2$ + 3H$_2$ ⇌ CH$_3$OH + H$_2$O), a highly relevant platform chemical and energy vector[1–3]. These systems are particularly attractive owing to their ability to suppress undesired carbon monoxide (CO) formation via the reverse water-gas shift (RWGS) reaction (CO$_2$ + H$_2$ ⇌ CO + H$_2$O), thereby leading to high methanol selectivity[2–4]. Additionally, ZnZrO$_x$ catalysts display remarkable on-stream stability even when exposed to impurities such as hydrogen sulfide and sulfur dioxide, which are commonly found in CO$_2$-containing streams[2,5]. Although coprecipitation (CP) was established as the first method for synthesizing ZnZrO$_x$ catalysts[1–3], a recent study revealed that flame spray pyrolysis (FSP)

yields materials with around threefold higher methanol space-time yield compared to coprecipitated systems[4]. Despite CP and FSP materials possessing similar catalytic ensembles, FSP maximizes surface area and formation of isolated Zn$^{2+}$ species located at surface positions of the ZrO$_2$ lattice, which foster enhanced performance[4]. Indeed, the unique architecture of the flame-made catalysts promotes the creation of surface oxygen vacancies, which are key components of the active ensembles that favor methanol formation. Regardless of the preparation method, however, the heterolytic activation of H$_2$ on ZnZrO$_x$ is the most demanding step and limits methanol formation rates[2–4,6–9].

Metal promotion is a well-established strategy for enhancing the hydrogen splitting ability and overall performance of oxide catalysts in

[1]Institute of Chemical and Bioengineering, Department of Chemistry and Applied Biosciences, ETH Zurich, Vladimir-Prelog-Weg 1, 8093 Zurich, Switzerland. [2]Institute of Chemical Research of Catalonia (ICIQ-CERCA), The Barcelona Institute of Science and Technology, Av. Països Catalans 16, 43007 Tarragona, Spain. [3]Laboratory of Physical Chemistry, Department of Chemistry and Applied Biosciences, ETH Zurich, Vladimir-Prelog-Weg 2, 8093 Zurich, Switzerland. [4]Department of Inorganic Chemistry and Materials Institute (IUMA), Faculty of Sciences, University of Alicante, Ap. 99, E-03080 Alicante, Spain. [5]Laboratory of Inorganic Chemistry, Department of Chemistry and Applied Biosciences, ETH Zurich, Vladimir-Prelog-Weg 1, 8093 Zurich, Switzerland. ✉e-mail: jpr@chem.ethz.ch

the conversion of $CO_2$ to methanol[10–16]. While coprecipitated $ZnZrO_x$ catalysts are known to benefit from the addition of small quantities of hydrogenation metals, examples are limited to a few metals ($M$ = Pd, Pt, and Cu), with palladium recognized as the most effective[6–9]. However, applying this strategy to flame-made $ZnZrO_x$ systems to improve their prospects for industrial implementation is challenging due to limited knowledge and transferability of promotional effects from conventionally synthesized catalysts. The effects of a given metal not only depend on its identity but also on its speciation, which is influenced by various factors, including the synthesis approach, promoter content, catalyst reconstruction under operating conditions, and structure of oxides used[6–10,12–14,17,18]. Despite substantial progress in understanding promotional effects on other oxide catalysts, like indium oxide-based systems[10–12,16], integrated studies of $ZnZrO_x$ catalysts are lacking. Therefore, the exploration of the promotion of flame-made $ZnZrO_x$ catalysts to elucidate the active-ensemble structures and associated performance of the resulting systems, including promoter speciation, impact on oxygen vacancy formation, and dynamics under reaction conditions are essential. In particular, operando investigations are key for establishing robust structure-performance relationships to guide the design of practical $M$-$ZnZrO_x$ catalysts[19–21].

In this study, we systematically investigate the promotion of flame-made $ZnZrO_x$ catalysts by relevant hydrogenation metals (0.5 mol% Re, Co, Au, Ni, Rh, Ag, Ir, Ru, Pt, Pd, and Cu) for $CO_2$ hydrogenation to methanol. Standardized catalyst synthesis and evaluation as well as in-depth characterization reveal copper as the most effective promoter for $ZnZrO_x$, as it leads to the largest performance improvement. In addition, Cu-promoted $ZnZrO_x$ catalysts comprise an earth abundant and cost-effective alternative to conventional palladium promotion and are compositionally distinct from traditional Cu-ZnO-$ZrO_2$ systems, which generally contain 20–40 mol% of Cu and are significantly less selective to methanol[1]. Detailed microscopy, kinetic, stability, operando spectroscopy analyses, and theoretical simulations are applied to gather understanding of copper promotion. Specifically, we investigate the speciation under reaction conditions and its effect on oxygen vacancy formation, the architecture of active ensembles, and reactivity. Our study provides valuable insights into $ZnZrO_x$ promotion by different metals, and offers unprecedented atomic-level rationalization of the working state of Cu-$ZnZrO_x$ catalysts for $CO_2$ hydrogenation to methanol.

## Results and discussion
### Impact of promoters on performance and catalyst architecture
Metal-promoted $ZnZrO_x$ catalysts were synthesized by flame spray pyrolysis (FSP, Supplementary Table 1), which was previously demonstrated to produce $ZnZrO_x$ materials with better catalytic performance than coprecipitated analogues for $CO_2$ hydrogenation to

methanol[4]. Accordingly, the optimal zinc content (5 mol%) reported was selected for this study, whereas a nominal amount of 0.5 mol% was chosen for all metal promoters (Re, Co, Au, Ni, Rh, Ag, Ir, Ru, Pt, Pd, and Cu), to produce catalysts with similar composition and thus enable a direct comparison of their promotional effects on $ZnZrO_x$. Both experimental compositions closely matched nominal values, as determined by inductively coupled plasma optical emission spectroscopy (ICP-OES, Supplementary Table 2). Performance evaluation at relevant $CO_2$ hydrogenation to methanol conditions ($T$ = 573 K, $P$ = 5 MPa, $GHSV$ = 24,000 $cm^3$ $h^{-1}$ $g_{cat}^{-1}$, and $H_2/CO_2$ = 4) showed that, for most $M$-$ZnZrO_x$ systems, methanol space-time yield ($STY$) remains virtually unchanged (Re- and Co-$ZnZrO_x$), slightly increased (ca. 10–20%, Au-, Ni-, Rh-, Ag-, and Ir-$ZnZrO_x$), or moderately improved (ca. 40–60%, Ru- and Pt-$ZnZrO_x$) in comparison to unpromoted $ZnZrO_x$ ($STY$ = 0.25 $g_{MeOH}$ $h^{-1}$ $g_{cat}^{-1}$, Fig. 1). This trend in methanol productivity can be traced back to the specific trade-off between $CO_2$ conversion ($X_{CO2}$) and methanol selectivity ($S_{MeOH}$) displayed by these catalysts, with Ru- and Pt-$ZnZrO_x$ attaining higher $X_{CO2}$ (ca. 6%) without experiencing significant loss in $S_{MeOH}$ (Supplementary Fig. 1). Remarkably, Cu exerts the highest improvement in the methanol $STY$ (ca. 120%, $STY$ = 0.55 $g_{MeOH}$ $h^{-1}$ $g_{cat}^{-1}$, Fig. 1), which surpasses that induced by Pd, the current state-of-the-art promoter for $ZnZrO_x$[6,9]. The striking methanol productivity of Cu-$ZnZrO_x$ is linked to its ability to sustain an outstanding $S_{MeOH}$ (ca. 90%) at high $X_{CO2}$ levels (ca. 9% Supplementary Fig. 1), a similar behavior also observed for Pd-$ZnZrO_x$. Interestingly, unpromoted and $M$-$ZnZrO_x$ catalysts show very similar specific surface area ($S_{BET}$) in fresh form (ca. 100 $m^2$ $g_{cat}^{-1}$, Supplementary Table 3 and Supplementary Fig. 2) which remains virtually unaltered after 20 h equilibration on stream. This indicates that these systems are particularly stable under reaction conditions and their performance difference does not stem from differences in $S_{BET}$. In addition, this highlights the advantages of using FSP, as it allows the reproducible synthesis of catalysts of varying compositions that can be used and compared without any pre-treatment (washing or calcination) and having to change the manufacturing procedure.

In-depth characterization was conducted to shed light on the architecture of $M$-$ZnZrO_x$ catalysts. Analysis by X-ray diffraction (XRD, Supplementary Fig. 3) revealed no reflections characteristic of pure metal phases or zinc oxide (ZnO) in the used catalysts, indicating that the promoters and zinc remain well dispersed or form amorphous phases. Concerning the $ZrO_2$ carrier, tetragonal ($t$) $ZrO_2$ is the predominant crystalline phase present in all fresh materials, which partially transforms into the monoclinic ($m$) structure upon reaction. This transformation is likely triggered by water formed under $CO_2$ hydrogenation conditions[4,10]. The partial transformation of $t$-$ZrO_2$ into $m$-$ZrO_2$ is a well-documented phenomenon for zirconia-containing materials prepared by FSP[4,10]. In contrast, $ZnZrO_x$ systems reported

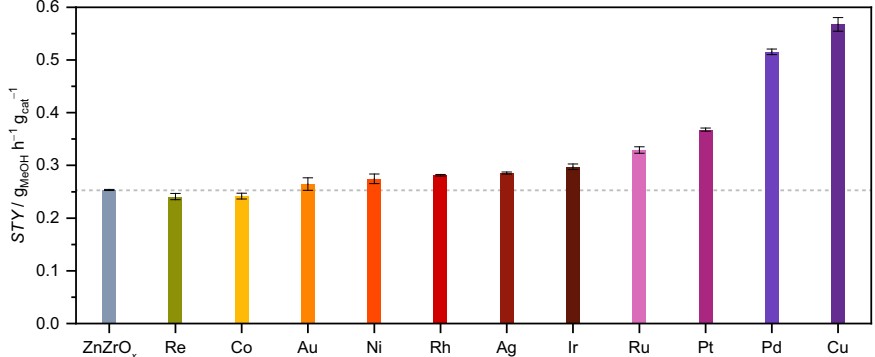

**Fig. 1 | Comparative performance of metal-promoted $ZnZrO_x$ catalysts.** Methanol space-time yield, $STY$ during $CO_2$ hydrogenation $M$-$ZnZrO_x$ (0.5 mol% of metal, $M$) catalysts prepared by FSP. The horizontal dashed line indicates the methanol $STY$ of the unpromoted $ZnZrO_x$ catalyst. Averaged values measured over 20 h on stream are presented with their corresponding error bars. Reaction conditions: $T$ = 573 K, $P$ = 5 MPa, $H_2/CO_2$ = 4, and $GHSV$ = 24,000 $cm^3$ $h^{-1}$ $g_{cat}^{-1}$.

in the literature are often prepared by coprecipitation, which leads to the incorporation of zinc into the bulk structure of $ZrO_2$, forming a solid solution that stabilizes the tetragonal phase, preventing its transformation into the monoclinic form. Comparatively, FSP typically results in materials with higher surface area and maximized zinc dispersion in the outmost layers of $ZrO_2$. This improved Zn utilization and leads to a less stabilized *t*-phase, which enables the thermodynamically favored *t*-to-*m* transformation, enhancing its $CO_2$ hydrogenation performance.

Investigations by scanning transmission electron microscopy (STEM) coupled to energy-dispersive X-ray (EDX) spectroscopy showed that metal nanoparticles with variable sizes (ranging from 5-10 nm) are clearly visible for used Au-, Ag-, Ru-, and Pd-$ZnZrO_x$ catalysts (Fig. 2 and Supplementary Fig. 4). For Au and Pd, these observations are corroborated by high-resolution STEM-HAADF analyses, which further revealed that, despite being undetected by EDX maps, nanoparticles of ca. 1 nm are present in Ir- and Pt-containing catalysts (Supplementary Fig. 5). In contrast, other metals like Re, Co, Ni, Rh, and Cu seem well dispersed as low-nuclearity species, as suggested by EDX maps, and, in the case of Re, also by STEM-HAADF (Fig. 2 and Supplementary Fig. 4,5). Interestingly, while zinc appears atomically dispersed over all catalysts, part of it tends to aggregate on Pd-$ZnZrO_x$ at regions containing palladium nanoparticles, suggesting that these phases are associated and likely form alloys. Temperature-programmed reduction with hydrogen ($H_2$-TPR, Supplementary Fig. 6) reveals that most $M$-$ZnZrO_x$ catalysts exhibit some distinguishable signals (<600 K) indicative of the reduction of promoters, particularly for Pd-$ZnZrO_x$, suggesting that pure metallic or Zn-containing alloys phases can be formed under reaction conditions[6,8,22].

X-ray absorption near-edge structure spectra (XANES) of $M$-$ZnZrO_x$ catalysts measured in quasi in situ mode (samples protected in capillaries under air-free environment) evidence differences in the metal promoters' speciation (Supplementary Fig. 7). Spectral features of the Ag- and Au-$ZnZrO_x$ systems probed at the Ag $K$ and Au $L_2$ edges, respectively, closely match those of metallic Ag and Au foils, indicating that Ag and Au form large metallic nanoparticles, in line with microscopy observations (Supplementary Fig. 4). In contrast, while promoters are in a metallic state in Pd-, Ru-, Rh-, Ir-, and Ni-$ZnZrO_x$ catalysts, they are likely interacting with zinc and possibly forming alloys, as suggested by the shift in their corresponding spectra compared to that of references. Still, the degree of alloying with zinc is low in several cases, as zinc mostly retains its oxidic character in all $M$-$ZnZrO_x$ catalysts (XANES, Zn $K$-edge, Supplementary Fig. 8). Indeed, $M$-Zn contributions can only be reliably assigned in the extended X-ray absorption fine structure spectra (EXAFS, Zn $K$-edge, Supplementary Fig. 9) of Pd-$ZnZrO_x$. Additionally, no signal characteristic of Zn-Zn scattering paths, generally centered at ca. 2.9 Å for ZnO, could be observed for any $M$-$ZnZrO_x$ catalyst, confirming the high dispersion of zinc species in most systems. Finally, the metals such as Pt, Re, Co, and Cu were mostly in oxidized state (Supplementary Fig. 7). In principle, since metals as cations are less prone to split $H_2$[23], this could rationalize

the limited performance of some of these elements as promoters. Still, it is particularly intriguing that oxidic copper species can effectively split $H_2$, which contrasts with its proven ability to improve the methanol productivity of $ZnZrO_x$ (Fig. 1), and therefore further investigation into its speciation under reaction conditions is required.

Overall, considering both microscopic (Fig. 2, and Supplementary Fig. 4,5) and spectroscopic (Supplementary Fig. 7-9) findings, no clear correlation between the speciation and the performance of $M$-$ZnZrO_x$ catalysts is observed, particularly when compared to $M$-$In_2O_3$ systems in which atomically-dispersed metal species are more effective promoters than clusters and nanoparticles[6,11]. Instead, promotion in $ZnZrO_x$ appears more metal-dependent than speciation-dependent, with copper and palladium displaying distinct speciation that lead to similar and superior performance. In the case of Pd, EXAFS fitting of the Pd $K$-edge spectra reveals a clear Pd-Zn contribution (ca. 6 Zn neighbors, Supplementary Fig. 10 and Supplementary Table 4), confirming the formation of PdZn nanoalloy particles, and in line with $H_2$-TPR (Supplementary Fig. 6). Since palladium is a well-known promoter for $ZnZrO_x$ and PdZn alloys are well documented for their favorable methanol synthesis properties[6,8,9,24], we have devoted special focus to gather fundamental understanding of $ZnZrO_x$ promotion by copper, which represents an earth-abundant, more sustainable, and unexplored promoter class.

## Copper speciation and promotional effect

To gain deeper insights into the origin of copper promotion in $ZnZrO_x$, Cu-$ZnZrO_x$ and corresponding reference Cu-$ZrO_x$ and $ZnZrO_x$ materials prepared by FSP were further investigated (Fig. 3a and Supplementary Fig. 11,12). Catalyst durability assessed in a 100-h test (Fig. 3a) confirms that Cu-$ZnZrO_x$ shows a significantly higher methanol productivity, which stabilizes after 50 h induction time ($STY$ = ca. 0.55 $g_{MeOH}$ $h^{-1}$ $g_{cat}^{-1}$) compared to that of $ZnZrO_x$ and Cu-$ZrO_x$ combined, indicating that its performance does not result from a simple additive effect of integrating both binary systems. The induction period observed for the Cu-$ZnZrO_x$ catalyst relates to the restructuring and equilibration of the catalyst at elevated temperature and pressure. A similar behavior is observed for the Cu-$ZrO_x$ catalyst, albeit requiring a shorter time (ca. 20 h), likely due to the less complex architecture of this catalyst. It is important to note that beyond the induction period, Cu-$ZnZrO_x$ demonstrates stable performance, greatly surpassing that of the Cu-$ZrO_x$ and $ZnZrO_x$ systems. The improved methanol $STY$ of Cu-$ZnZrO_x$ compared to $ZnZrO_x$ and Cu-$ZrO_x$ is linked to its higher $X_{CO2}$ (ca. 9 versus 4 and 3%, respectively) and, in the case of Cu-$ZrO_x$, also $S_{MeOH}$ (ca. 90 versus 70%), as evidenced by $S_{MeOH}$ evaluation carried out at similar $X_{CO2}$ levels (ca. 5%, Fig. 3b). For completeness, Cu-$ZnO_x$ was also synthesized and its performance evaluated (Supplementary Fig. 11), which is comparable to Cu-$ZrO_x$ but notably inferior to Cu-$ZnZrO_x$, further highlighting the superior catalytic properties of the ternary system. Interestingly, while all catalysts exhibit a low apparent reaction order with respect to $H_2$ ($n_{H2}$) for methanol synthesis (Fig. 3c), Cu-$ZnZrO_x$ displays a negative $n_{H2}$ for the RWGS reaction,

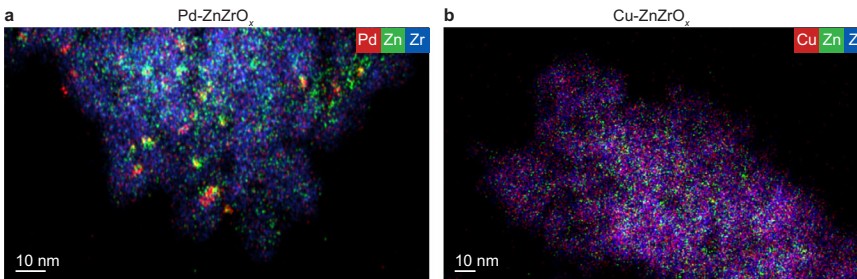

**Fig. 2 | Structural organization of Pd- and Cu-$ZnZrO_x$ catalysts.** EDX maps of (**a**) Pd-$ZnZrO_x$ and (**b**) Cu-$ZnZrO_x$ catalysts after $CO_2$ hydrogenation for 20 h. Reaction conditions as in the caption of Fig. 1.

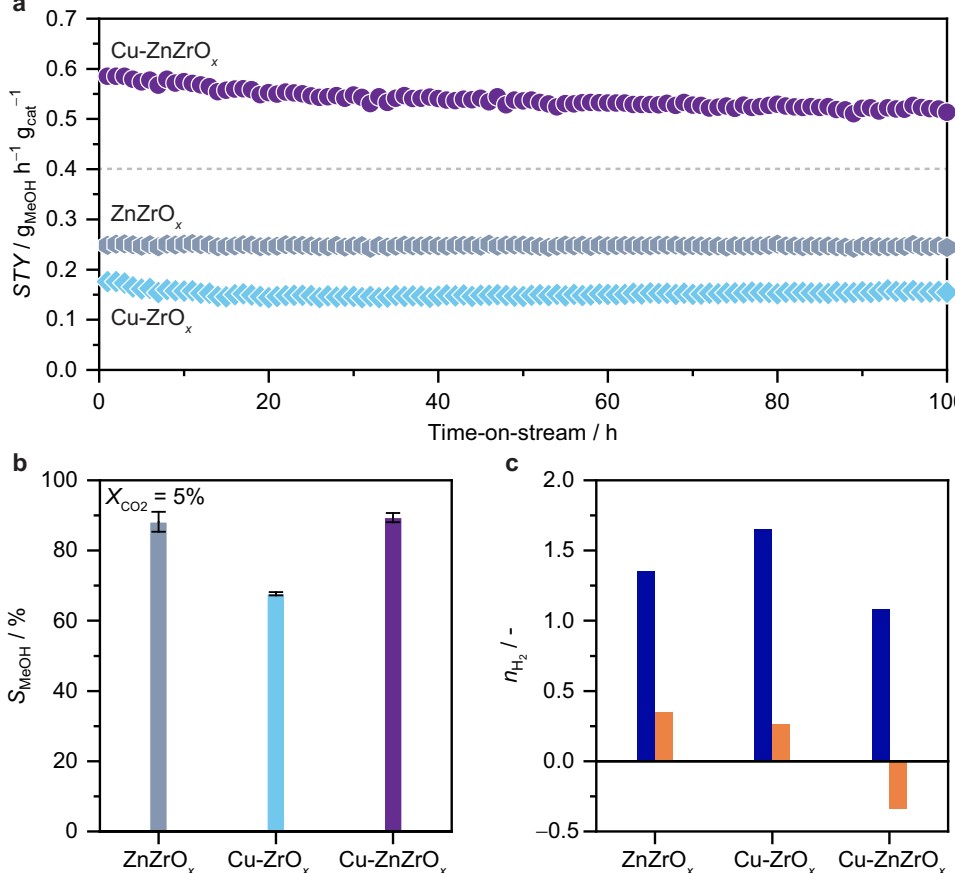

**Fig. 3 | Stability and kinetics of the Cu-ZnZrO$_x$ catalyst. a** Methanol space-time yield, *STY*, (**b**) selectivity, $S_{MeOH}$ during CO$_2$ hydrogenation over ZnZrO$_x$, Cu-ZrO$_x$, and Cu-ZnZrO$_x$ catalysts. (**c**) Apparent reaction order with respect to H$_2$ ($n_{H2}$) for CO$_2$ hydrogenation to methanol (blue) and the RWGS reaction (orange). The horizontal line indicates the sum of the methanol *STY* of ZnZrO$_x$ and Cu-ZrO$_x$ catalysts.

$S_{MeOH}$ was assessed at constant CO$_2$ conversion (ca. 5%) by adjusting the *GHSV* (24,000–96,000 cm$^3$ h$^{-1}$ g$_{cat}^{-1}$). Averaged values measured over 20 h on stream are presented with their corresponding error bars. Reaction conditions as in the caption of Fig. 1.

indicating that CO formation is inhibited over Cu-ZnZrO$_x$ at the applied hydrogenation conditions, rationalizing its high methanol selectivity. These findings are in line with copper agglomerating into nanoparticles in Cu-ZrO$_x$ upon reaction, which is known to favor CO formation[25–27] and, although not detected by XRD (Supplementary Fig. 13), it is clearly evidenced by STEM-EDX (Supplementary Fig. 14a). In contrast, copper remains well dispersed as low-nuclearity species in Cu-ZnZrO$_x$ (Supplementary Fig. 14b), as copper atoms are most likely associated with zinc, indicating that this speciation favors methanol formation over CO production.

To shed light into the copper speciation in the Cu-ZnZrO$_x$ catalyst at work, operando X-ray absorption spectroscopy (XAS) experiments were conducted under pretreatment in He (1.5 MPa and from 303 to 573 K) and CO$_2$ hydrogenation conditions (1.5 MPa and 573 K) at the Cu and Zn *K*-edges. Operando XANES reveals that both copper and zinc exist as oxidized phases in the fresh Cu-ZnZrO$_x$ catalyst (Fig. 4a, b). Upon thermal treatment in He, both elements undergo partial reduction, likely forming CuO$_x$ and ZnO$_x$ species. Interestingly, switching to the reaction mixture causes abrupt changes to copper, and more metallic character is evidenced, which contrasts the quasi in situ findings that show oxidized copper species after reaction (Fig. 4a and Supplementary Fig. 8). This apparent discrepancy highlights the importance of operando studies and can be attributed to underlying phenomena such as the highly oxophilic character of copper, which, in the absence of a reducing environment, can promote its reoxidation even by small concentration of oxygen impurities typically present in He streams used for preserving samples before analysis. Still, spectral

features of Cu-ZnZrO$_x$ under operating conditions deviate from those of Cu-ZrO$_x$, which contains metallic Cu nanoparticles (Fig. 4a). This hints at an effective stabilization of copper as low-nuclearity metallic species on the mixed ZnZrO$_x$ surface, owing most likely to a strong interaction with zinc. Nonetheless, while zinc shows further reduction when exposed to the reaction mixture (Fig. 4b), it is not fully reduced and still maintains characteristic features of ZnO$_x$, suggesting that only a small fraction of metallic zinc atoms forms and interacts with copper, while its majority may comprise partially reduced zinc species intermixed on the ZrO$_2$ surface. This rapidly evolving architecture remains unchanged after 5 h on stream (Fig. 4a, b), accounting for the stable performance of Cu-ZnZrO$_x$ (Fig. 3a).

In line with XANES findings, EXAFS analysis of the fresh Cu-ZnZrO$_x$ and Cu-ZrO$_x$ catalysts exhibits strong metal-oxygen (*M*-O) interactions, with ca. 4 oxygen neighbors (Fig. 4c and Supplementary Table 4). Under the highly reducing CO$_2$ hydrogenation environment, however, a clear difference is observed between these two catalysts. Specifically, Cu-Cu contributions similar to those of Cu foil are present on Cu-ZrO$_x$, with a large number of copper neighbors (ca. 9, Supplementary Table 4), which is consistent with the formation of metallic extended surfaces or nanoparticles. In contrast, while Cu-*M* signals are evidenced for Cu-ZnZrO$_x$, they are clearly shifted from Cu-Cu foil but distinguishing between Cu-Cu and Cu-Zn interactions is impossible due to their similar scattering factors (Fig. 4c). Regarding the Zn *K*-edge, the EXAFS spectra shows that Cu-ZnZrO$_x$ largely exhibits Zn-O interactions and only a small Zn-*M* contribution is observed, which is shifted compared to that of the Zn foil (Fig. 4c). Together with XANES

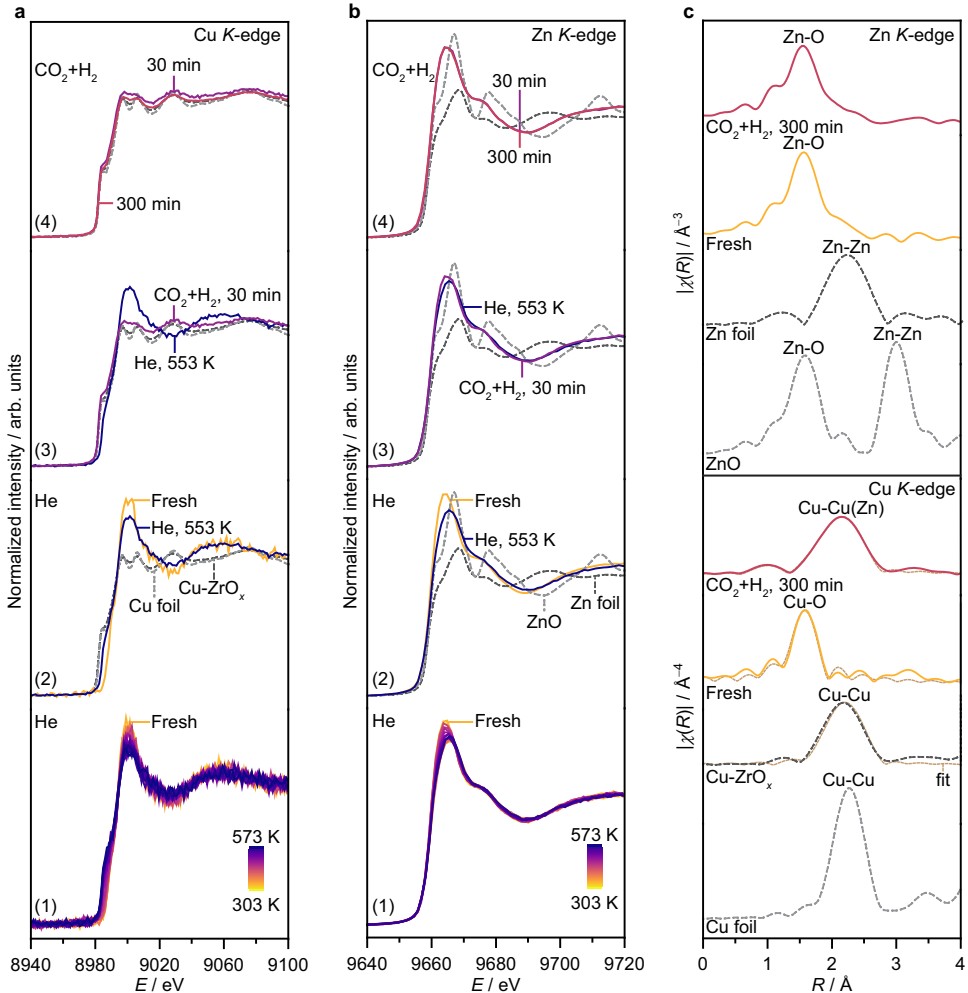

**Fig. 4 | Copper and zinc speciation in the Cu-ZnZrO$_x$ catalyst.** *Operando* (**a**) Cu and (**b**) Zn K-edge XANES spectra of Cu-ZnZrO$_x$ catalyst (**1**) during the heating ramp in He ($m_{cat}$ = 0.013 g, heating rate = 5 K min$^{-1}$, $T$ = 573 K, $P$ = 1.5 MPa, and dwell time = 30 min), (**2**) under He at 573 K, and under reaction conditions ($m_{cat}$ = 0.013 g, $F_T$ = 15 cm$^3$ min$^{-1}$, $T$ = 573 K, $P$ = 1.5 MPa, H$_2$/CO$_2$ = 4, dwell time = 300 min) after (**3**) 30 min

and (**4**) 300 min on stream. (**c**) Fourier-transformed EXAFS spectra of Cu-ZnZrO$_x$ catalyst in fresh form and under reaction conditions corresponding to the spectra in (**a**) and (**b**). XANES and EXAFS spectra of Cu and Zn foils, ZnO, and activated Cu-ZrO$_x$ ($m_{cat}$ = 0.013 g, $F_T$ = 15 cm$^3$ min$^{-1}$, $T$ = 573 K, $P$ = 1.5 MPa, H$_2$/CO$_2$ = 4, dwell time = 300 min) are shown as reference.

and STEM-EDX results (Figs. 2 and 4a), these observations indicate that copper is in a metallic state and strongly associated with partially reduced zinc atoms, most likely forming low-nuclearity zinc-rich CuZn clusters, particularly since the overall zinc content is 10 times higher than that of copper and the latter element is highly dispersed.

Density functional theory (DFT) simulations were performed to rationalize the most likely architectures present on ZnZrO$_x$, Cu-ZrO$_x$, and Cu-ZnZrO$_x$ catalysts. For this purpose, more than 200 structures were examined to evaluate the degree of incorporation of Cu and Zn into *m*-ZrO$_2$ surface sites, formation of oxygen vacancies, Cu adsorption trends, and interaction between Cu and Zn in the ZrO$_2$ matrix. Models were built based on two criteria: (i) the compatibility with the experimental observations (cluster nature); and (ii) the DFT-computed energies were reasonable (i.e., processes thermodynamically favored). The most relevant models and their corresponding potential energies (*E*) are depicted in Fig. 5, whereas other models are discussed in the Supplementary Information (see Supplementary Methods and Discussion). ZnZrO$_x$ is best represented by a model with isolated Zn atoms replacing Zr sites at the ZrO$_2$ surface with a nearby oxygen vacancy, (ZnZrO$_x$, $\Delta E$ = −0.04 eV, Fig. 5 and Supplementary Fig. 15)[4]. In contrast, two independent models, *i.e.* Cu(111) and *m*-ZrO$_2$(−111), are considered for the Cu-ZrO$_x$ catalyst, particularly since experimental results evidence two separated phases (Supplementary Fig. 14a), and both

incorporation and adsorption of a Cu atom on ZrO$_2$ are endothermic (Cu@ZrO$_x$, $\Delta E$ = 1.86 eV, Cu/ZrO$_2$, $\Delta E$ = 2.63 eV, respectively, see Fig. 5). For CuZnZrO$_x$, we initially explored the incorporation of Cu in the ZnZrO$_x$ model with isolated Zn centers (Fig. 5). Cu incorporation (Cu@Zn$_4$ZrO$_x$) is not favored ($\Delta E$ = 2.08 eV wrt bulk Cu) even at sites nearby Zn atoms. Interestingly, adsorption of Cu on the surface near Zn sites (Cu/Zn$_4$ZrO$_x$, Fig. 5) is favored compared to zinc-free ZrO$_2$, further confirming the Cu-Zn affinity. Nevertheless, none of the configurations investigated with separated Zn atoms provide suitable nucleation sites for Cu (Fig. 5). Additionally, none of the models fully agree with the EXAFS and STEM-EDX findings, which indicate that Cu is most likely surrounded by a zinc-rich ensemble under reaction conditions (Supplementary Table 4). However, the formation of CuZn clusters under reaction conditions should be traced back to the Cu-Zn affinity, since the formation energy of the bulk Cu$_5$Zn$_8$ (brass) and CuZn alloys are exothermic ($E_{Cu5Zn8}$ = −0.26 eV and $E_{CuZn}$ = −0.17 eV, respectively). Hence, alternative scenarios include that the formation of the Cu-Zn pairs occurs earlier in the preparation. Indeed, the melting temperatures of Cu-Zn and ZrO$_2$ differ significantly (1023 and 2988 K) meaning that the nucleation during cooling of the FSP could occur independently. This would lead to Zn rich environments with Cu and later the stabilization of these clusters by the oxidic ZrO$_2$ phase ensuring that some O atoms bond the oxophilic Zn ones. Thus, we

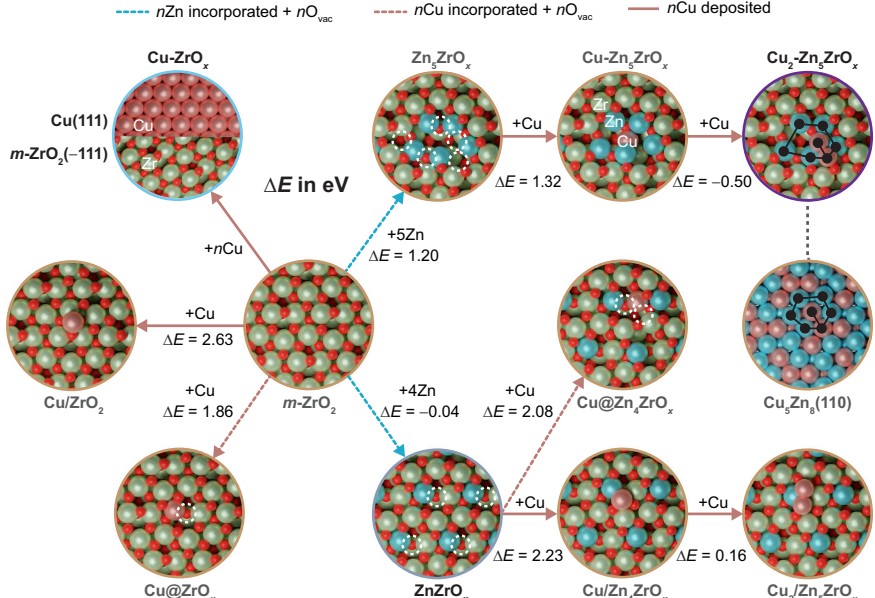

**Fig. 5 | Theoretical description of copper and zinc sites on the Cu-ZnZrO$_x$ catalyst.** Snapshots of DFT models with their associated relative potential energy, $\Delta E$ with respect to Cu and Zn bulk energies and $m$-ZrO$_2$ surface, were employed to rationalize the tendency of Zn and Cu to incorporate into the ZrO$_2$ surface sites and the deposition of Cu on Zn-doped and pure ZrO$_2$ surfaces. The models employed to assess Cu-ZrO$_x$, ZnZrO$_x$, and Cu-ZnZrO$_x$ catalytic systems via DFT simulations are highlighted in light blue, gray, and purple, respectively. The dashed arrows represent the incorporation of $n$ Zn or Cu atoms (blue and light pink, respectively) accompanied with the formation of $n$ oxygen vacancies on the ZrO$_2$ surface. Solid light pink lines denote the deposition of a Cu atom on ZrO$_2$ and ZnZrO$_x$ surfaces. The common Zn$_5$Cu$_2$ pattern found in Cu$_2$-Zn$_5$ZrO$_x$ and Cu$_5$Zn$_8$(110) is depicted in black. Color code of DFT models: Zn (blue), Zr (green), Cu (light pink), O (red), and O vacancies (doted white circles).

investigated models where 5 Zn atoms aggregate in nearby sites (Zn$_5$ZrO$_x$) within the $m$-ZrO$_2$ surface with 5 oxygen vacancies. We employed this structure to study the adsorption of a single Cu atom (Cu-Zn$_5$ZrO$_x$) finding that this is more favored than Cu/ZrO$_2$ and Cu/ Zn$_4$ZrO$_x$ (Fig. 5). Including a second Cu atom is exothermic (Cu$_2$-Zn$_5$ZrO$_x$) and results in a CuZn motif very similar to that found in brass, particularly the Cu$_5$Zn$_8$ (110) surface (Fig. 5). Moreover, larger Cu clusters (containing 3 or 4 atoms) are not favored (Supplementary Fig. 16) further underlying the role of nucleation and alloy properties in the final compound. Furthermore, alternative models (over 150 structures) were evaluated to explore the robustness of the Cu$_2$-Zn$_5$ZrO$_x$ ensemble. We assessed different degrees of reduction and the formation of analogous structures with different amount of Zn and Cu, as detailed in the Supplementary Information (see Supplementary Methods and Discussion, Supplementary Fig. 17-21, and Supplementary Table 5). These results indicate that models with 2 copper (possibly forming dimer species) and 5 zinc atoms are more likely thermodynamically and the most compatible with the experimental observations. Therefore, the Cu$_2$-Zn$_5$ZrO$_x$ structure was selected as the most representative model for evaluating the catalytic properties of Cu-ZnZrO$_x$ catalysts from a computational standpoint.

## Mechanistic insights into oxygen vacancy formation and dynamics

To investigate the formation and dynamics of oxygen vacancies on Cu-ZnZrO$_x$, in situ electron paramagnetic resonance spectroscopy (EPR) experiments were carried out on this catalyst and reference ZnZrO$_x$ and Cu-ZrO$_x$ materials (Fig. 6). Fresh ZnZrO$_x$ shows a weak and almost isotropic peak ($g = 2.003$) and an additional anisotropic signal ($g_\parallel = 1.959$, $g_\perp = 1.977$, Fig. 6a), which are attributed to unpaired electrons trapped in isolated oxygen vacancies (behaving paramagnetically, $V_O^-$-p) and Zr$^{3+}$ sites, respectively, as previously reported[4]. Upon thermal treatment in He, the intensity of both signals increased (Fig. 6a), indicating the formation of a relatively small amount of thermally-induced vacancies. In contrast, exposure to the reaction mixture (CO$_2$ + H$_2$) led to a decrease in the Zr$^{3+}$ signal (Fig. 6a and Supplementary Fig. 22a). This is consistent with previous observations indicating that Zr$^{3+}$ is reoxidized to Zr$^{4+}$ as electrons are transferred to vacancies ($V_O^{2-}$) formed by reaction with H$_2$ (see Supplementary Discussion for detailed description)[4]. Still, Zr$^{3+}$ does not effectively impact the catalytic ensemble properties, acting as a spectator species[4,10]. Upon switching to reaction mixture a newly broad signal was evidenced and attributed to unpaired electrons delocalized over several oxygen vacancies that interact (exchange-coupled vacancy polarons, $V_O^-$-f)[4] (Fig. 6a and Supplementary Fig. 22a). Generally, this broad signal appears when the density of vacancies becomes particularly high, which is considered a key performance descriptor, as reported for Pd-In$_2$O$_3$-ZrO$_2$ systems[10]. For Cu-ZrO$_x$, however, only Cu$^{2+}$ and $V_O^-$-p species are detected under thermal treatment in He, whereas Zr$^{3+}$ and $V_O^-$-f signals are not observed (Fig. 6b and Supplementary Fig. 22b) even under CO$_2$ + H$_2$, suggesting that a low concentration of vacancies is present on Cu-ZrO$_x$, which could in principle explain its inferior performance (Fig. 3a).

The EPR spectrum of fresh Cu-ZnZrO$_x$ displays a typical Cu$^{2+}$ signal, with an axial $g$ tensor and a well-resolved hyperfine splitting of the $g_\parallel$ component (Fig. 6c). Interestingly, alike Cu-ZrO$_x$, no Zr$^{3+}$ signal was observed, indicating that Cu$^{2+}$ sites are better electron scavengers. Thermal treatment in He led to a strong decrease of the Cu$^{2+}$ and increase of $V_O^-$-p signals, while a new feature appeared around $g = 2$, partially overlapped with the Cu$^{2+}$, and consistent with that of surface-adsorbed superoxide ions (Fig. 6c). However, contrary to ZnZrO$_x$, electrons released during these processes are most likely trapped by (i) Cu$^{2+}$ sites, generating EPR-silent Cu$^+$ species, (ii) empty vacancies, giving rise to $V_O^-$-p, and (iii) oxygen released from the surface lattice, leading to the formation of the superoxide. This further corroborates its presence significantly promotes vacancy formation. Exposure to the reaction mixture leads to the complete reduction of Cu$^{2+}$ and to a significant decrease of the $V_O^-$-p signal (Fig. 6c, d). Additionally, a signal assigned to $V_O^-$-f is detected but much broader than the one observed for ZnZrO$_x$, indicating that a higher density of vacancies is

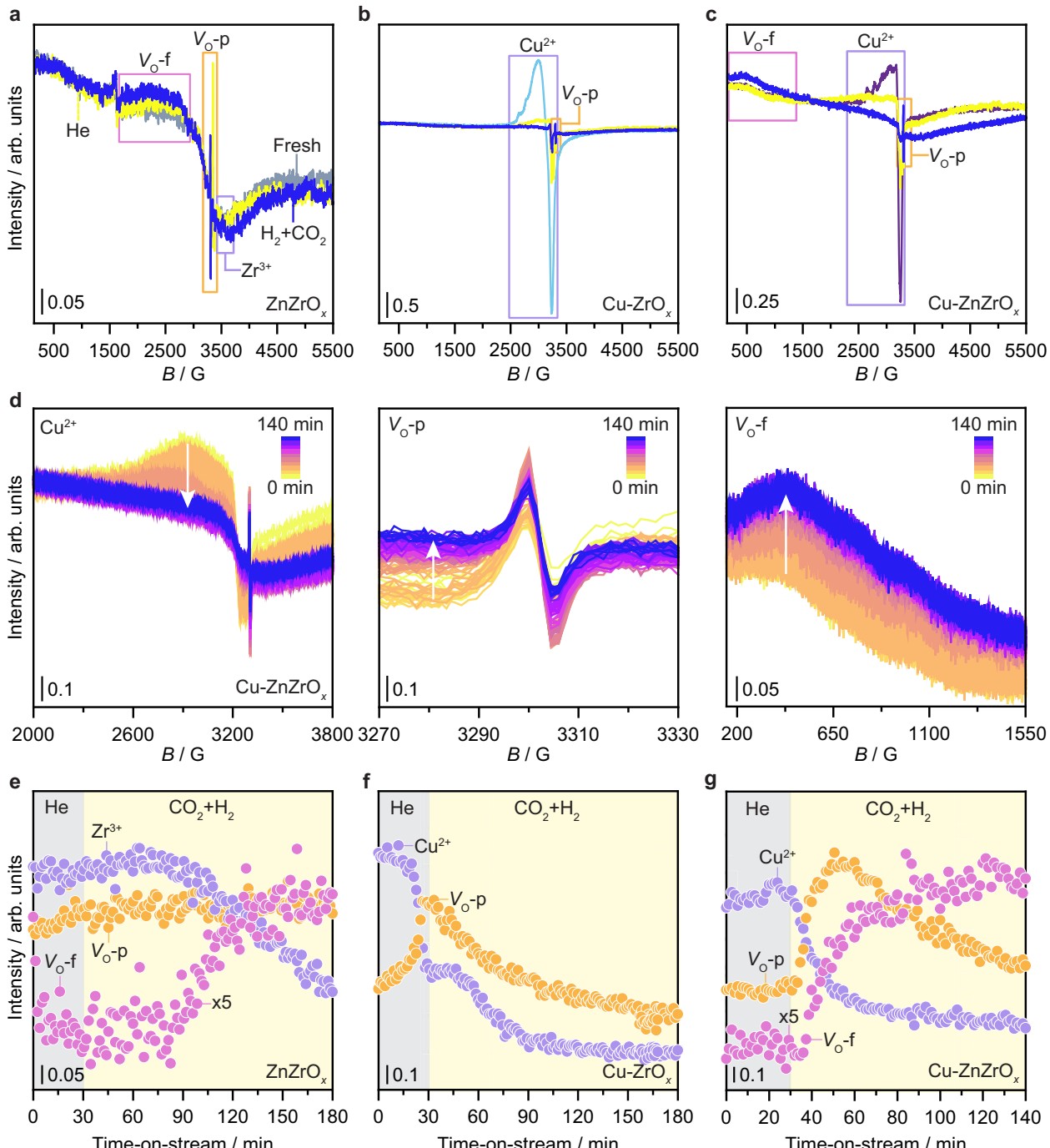

**Fig. 6 | Oxygen vacancy dynamics during CO₂ hydrogenation.** In situ EPR spectra measured at room temperature of (**a**) $ZnZrO_x$, (**b**) $Cu\text{-}ZrO_x$, and (**c**) $Cu\text{-}ZnZrO_x$ catalysts in fresh form, after heating ramp in He ($m_{cat}$ = 0.013 g, $F_T$ = 15 cm³ min⁻¹, heating rate = 5 K min⁻¹, $T$ = 573 K, $P$ = 1 MPa) and CO₂ hydrogenation ($m_{cat}$ = 0.013 g, $F_T$ = 15 cm³ min⁻¹, $T$ = 573 K, $P$ = 1 MPa, and H₂/CO₂ = 4). (**d**) Operando EPR spectra of key signals for $Cu\text{-}ZnZrO_x$ catalyst under He ($m_{cat}$ = 0.013 g, $F_T$ = 15 cm³ min⁻¹, $T$ = 573 K, $P$ = 1 MPa, and dwell time = 30 min) and under reaction conditions ($m_{cat}$ = 0.013 g, $F_T$ = 15 cm³ min⁻¹, $T$ = 573 K, $P$ = 1 MPa, and H₂/CO₂ = 4) with continuous time on stream. Evolution of $Cu^{2+}$ and $Zr^{3+}$ species, as well as paramagnetic and ferromagnetic oxygen vacancies ($V_O^-$-p and $V_O^-$-f, respectively) for (**e**) $ZnZrO_x$, (**f**) $Cu\text{-}ZrO_x$, and (**g**) $Cu\text{-}ZnZrO_x$ catalysts under operando conditions described in **d** and Supplementary Fig. 22a,b.

formed on $Cu\text{-}ZnZrO_x$. The decrease of the $V_O^-$-p signal is most likely linked to its conversion into the broad feature ($V_O^-$-f), as the increased concentration of vacancies leads to ferromagnetically-coupled defect sites and, consequently, depletes the number of isolated vacancies ($V_O^-$-p).

To gather additional insights into the dynamics of oxygen vacancies on $Cu\text{-}ZnZrO_x$ catalyst, operando EPR measurements were performed under He and CO₂ hydrogenation conditions (1 MPa and

573 K; Fig. 6d–g, Supplementary Fig. 22a, b, and 23). Evolution of key species over time (Fig. 6e–g) reveals that $V_O^-$-p and $V_O^-$-f formation rates are mirrored by that of $Cu^{2+}$ and $Zr^{4+}$ reduction, suggesting that these processes are linked, thereby validating the above-discussed electron scavenging mechanisms. More importantly, these experiments show that $V_O^-$-f signals are better fingerprints compared to $V_O^-$-p for monitoring vacancy formation and annihilation (see Supplementary Discussion for details). Additionally, a comparison of the $Cu^{2+}$

reduction kinetics of Cu-ZnZrO$_x$ and Cu-ZrO$_x$ shows significant differences (Supplementary Fig. 24), with a monotonous mono-exponential decay being observed for Cu-ZnZrO$_x$. In contrast, Cu-ZrO$_x$ clearly shows a component that decays with the same rate as the Cu$^{2+}$ in Cu-ZnZrO$_x$, whereas the other starts decreasing at a later stage and is significantly slower, which is most likely related to the reduction of copper nanoparticles present on Cu-ZrO$_x$, in line with microscopy results (Supplementary Fig. 14a). Furthermore, these observations indicate that the overall Cu$^{2+}$ reduction is more efficient on Cu-ZnZrO$_x$ due likely to a series of coupled factors such as its speciation and strong interaction with zinc and oxygen vacancies. Interestingly, exposing the Cu-ZnZrO$_x$ catalyst to oxygen at room temperature after the reaction reveals a full recovery of the Cu$^{2+}$ signal (Supplementary Fig. 25), hinting at the stable nature of copper sites. Overall, EPR experiments demonstrate that oxygen vacancy generation is augmented on Cu-ZnZrO$_x$ upon reaction, which most likely contributes to its enhanced performance. Indeed, these findings strongly suggest that these defect sites are an integral component of the low-nuclearity zinc-rich CuZn catalytic ensembles present on Cu-ZnZrO$_x$, with their role being multifaceted and likely including (i) stabilization of copper atoms and/or reaction intermediates, and (ii) activation of reactants.

## Insights on the reaction mechanism

CO$_2$ hydrogenation to methanol requires the efficient activation of CO$_2$ and H$_2$, followed by selective proton-hydride transfers to avoid the CO formation through the RWGS reaction. A bifunctional mechanism combining the properties of oxidic (CO$_2$ activation and hydrogenation) and metallic (H$_2$ activation) phases in close proximity is key to avoid transport of active species and achieve high methanol productivities[11,19]. Accordingly, adsorption of key reactants, intermediates, and byproduct species (CO$_2$, CO, H$_2$, HCOO*, and CH$_3$O*) on selected models displayed in Fig. 5 was assessed through DFT simulations to rationalize the catalytic properties of Cu-ZrO$_x$, ZnZrO$_x$, and Cu-ZnZrO$_x$. CO$_2$ adsorption is favored on basic sites of $m$-ZrO$_2$ (−111) whereas H$_2$ activation on this surface is endothermic (Fig. 7), with an activation energy ($E_a$) of 0.70 eV, and occurs heterolytically, forming hydrides (H$^-$) and protons (H$^+$) on a Zr-O pair (Supplementary Fig. 26). In contrast, Cu (111) poorly activates CO$_2$ but homolytic H$_2$ dissociation takes place ($E_a$ = 0.58 eV) and the adsorption of its products (hydrogen atoms) is highly favored (Fig. 7). Hence, spillover from Cu nanoparticles onto the ZrO$_2$ surface, where CO$_2$ is activated and hydrogenated, is required for the Cu-ZrO$_x$ catalyst. Consequently, this leads

to a sluggish transport of active species, which hampers methanol synthesis on this system, thereby contributing to its inferior performance. For ZnZrO$_x$, heterolytic H$_2$ activation on a Zn-O pair is more favored ($E_a$ = 0.21 eV and $E_{ads,H2}$ = 0.02 eV) compared to ZrO$_2$ (Figs. 7 and 9). Additionally, CO$_2$ is activated as carbonate in a nearby site, thus avoiding unnecessary transport of active species onto the catalyst surface, which explains its higher methanol productivity compared to Cu-ZrO$_x$. Adsorption energies for CO$_2$ and H$_2$-dissociated products on the Cu$_2$-Zn$_5$ZrO$_x$ model, which better describes active ensembles on the Cu-ZnZrO$_x$ system, indicate that this configuration enhances activation of these reactants at spatially resolved sites, thereby also circumventing long-range transport of species (Fig. 7). Moreover, H$_2$ activation is barrierless and found to proceed in a homolytic manner on metallic Cu atoms (Figs. 7 and 9), further corroborating the key role of Cu in improving the H$_2$ splitting ability of ZnZrO$_x$. Similar results were observed for analog models also comprising partially reduced Zn atoms incorporated into surface lattice positions of ZrO$_2$, oxygen vacancies, and metallic Cu atoms deposited on a different arrangement (Cu$_2$-Zn$_5$ZrO$_x$-b, Supplementary Fig. 27). In contrast, adsorption energies of reactant species on the Cu$_5$Zn$_8$ (110) surface exhibit similar trends as Cu(111), favored H$_2$ activation and unfavored CO$_2$ adsorption, suggesting that partially reduced zinc species, present on Cu-ZnZrO$_x$, rather than metallic counterparts are required to enable an efficient activation and subsequent transformation of reactants into methanol (Supplementary Fig. 27).

To gain experimental insights into key reactive intermediate species formed on Cu-ZrO$_x$, ZnZrO$_x$, and Cu-ZnZrO$_x$ catalysts under reaction conditions, operando diffuse reflectance infrared Fourier transform spectroscopy (DRIFTS) experiments were conducted (Fig. 8 and Supplementary Fig. 28). Upon exposure to the reaction mixture (H$_2$/CO$_2$ = 4, 1.5 MPa, and 573 K; Fig. 8a-c, and Supplementary Fig. 28), all catalysts exhibited bands characteristic of formate (HCOO*, 1586, 1373, 2736, 2875, and 2965 cm$^{-1}$) and methoxy species (CH$_3$O*, 1051, 1140, 2827, and 2928 cm$^{-1}$), which can be assigned to distinct vibrational modes (Supplementary Table 6, 7)[2,25,28–30]. These results demonstrate that methanol synthesis via CO$_2$ hydrogenation over all catalysts follows the formate pathway, which is in line with previous studies on ZnZrO$_x$[2,6] and indicates that the copper promotion in Cu-ZnZrO$_x$ is not governed by a different mechanism. Interestingly, a strong signal attributed to gas-phase CO[6,31] (Supplementary Fig. 28) is detected for Cu-ZrO$_x$, which is in agreement with its tendency to promote this undesired product, thus explaining its lower $S_{MeOH}$

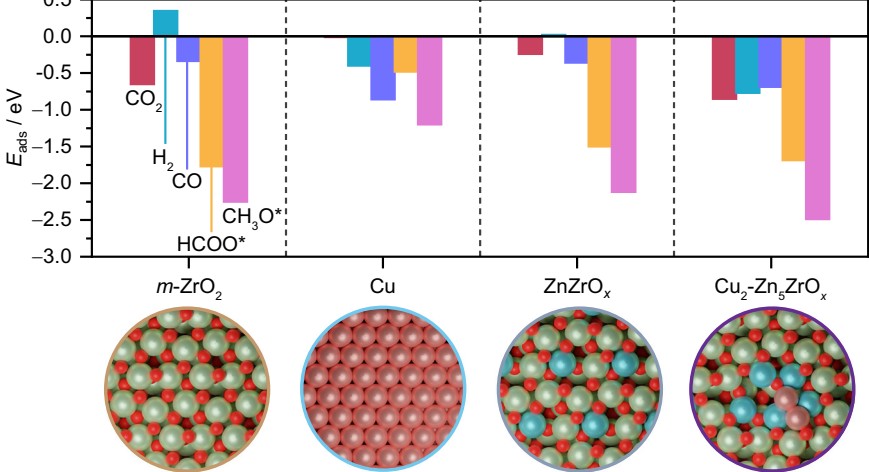

**Fig. 7 | Adsorption energies of reactive species.** Adsorption energies, $E_{ads}$ of key reactive species on relevant surfaces representative of Cu-ZrO$_x$, ZnZrO$_x$, and Cu-ZnZrO$_x$ catalytic systems. Color code of DFT models: Zn (blue), Zr (green), Cu (light pink), and O (red). While only the most energetically favorable $E_{ads}$ of key reactive species for each model is displayed in Fig. 7, other adsorption structures and $E_{ads}$ and are shown in Supplementary Fig. 26 and 27, respectively.

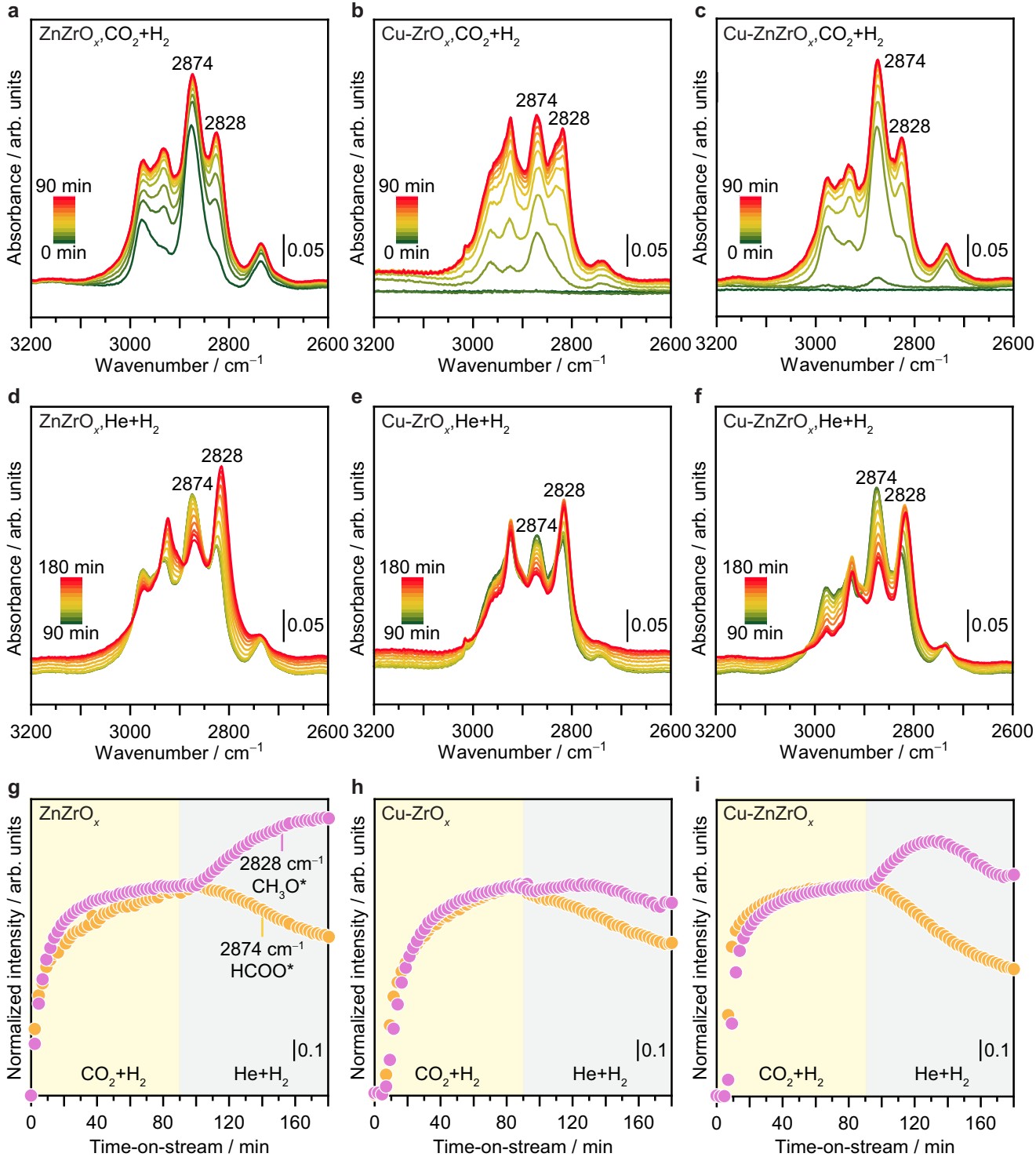

**Fig. 8 | Evolution of reaction intermediates.** Operando DRIFT spectra of key signals for (**a**) ZnZrO$_x$, (**b**) Cu-ZrO$_x$, and (**c**) Cu-ZnZrO$_x$ catalysts under (**a-c**) CO$_2$ hydrogenation ($F_T$ = 40 cm$^3$ min$^{-1}$, $T$ = 573 K, $P$ = 1.5 MPa, H$_2$/CO$_2$ = 4, and dwell time = 90 min) and (**d-f**) after switching to a mixture of H$_2$ balanced with He ($F_T$ = 40 cm$^3$ min$^{-1}$, $T$ = 573 K, $P$ = 1.5 MPa, H$_2$/He = 4, dwell time = 90 min). Evolution of evolution of normalized intensity of formate (HCOO*, 2874 cm$^{-1}$, orange) and methoxy (CH$_3$O*, 2828 cm$^{-1}$, purple) for (**g**) ZnZrO$_x$, (**h**) Cu-ZrO$_x$, and (**i**) Cu-ZnZrO$_x$ catalysts under operando conditions described in (**a–f**).

compared to ZnZrO$_x$ and Cu-ZnZrO$_x$ (Fig. 3b). Evolution of HCOO* and CH$_3$O* species assessed by monitoring the vibration bands at 2874 and 2828 cm$^{-1}$, respectively, over time-on-stream (Fig. 8g–i) revealed that their surface concentration changes simultaneously for all catalysts, suggesting that overall reaction rate is determined by the HCOO* surface concentration. Still, the surface concentration of HCOO* and CH$_3$O* on Cu-ZnZrO$_x$ reaches steady state within ca. 30 min, whereas it

requires ca. 50 and 70 min for ZnZrO$_x$ and Cu-ZrO$_x$, respectively (Fig. 8g–i). This suggests that the catalytic ensemble in Cu-ZnZrO$_x$ is more effective in accelerating the formation of these key reaction intermediates. To investigate the dynamics of surface species in pure H$_2$, CO$_2$ was replaced by He and the normalized intensity of the vibration bands at 2874 and 2828 cm$^{-1}$ was continuously monitored (Fig. 8d–i). The concentration of HCOO* decreases rapidly, gradually,

and slowly for Cu-ZnZrO$_x$, ZnZrO$_x$, and Cu-ZrO$_x$, respectively, over 90 min on stream, hinting at a much lower energy barrier for its hydrogenation to CH$_3$O* on the copper-promoted ZnZrO$_x$ catalyst (Fig. 8g–i). Still, unlike ZnZrO$_x$ and Cu-ZrO$_x$, Cu-ZnZrO$_x$ swiftly hydrogenates newly formed CH$_3$O* species, reaching a steady state similar to that in the presence of both CO$_2$ and H$_2$ (Fig. 8i). In this case, further hydrogenation of CH$_3$O* is most likely masked by the dissociative adsorption of methanol on unoccupied sites created by the absence of CO$_2$[32].

To gain further insights into the reaction mechanism, we computed energy profiles for CO$_2$ hydrogenation to methanol and CO on the relevant models presented in Fig. 5 (Supplementary Figs. 29, 30). We followed the formate pathway, which is in line with operando DRIFTS results and previous studies of ZnZrO$_x$[2,4,6] and other metal oxide-based catalysts[6,11,33]. We have also evaluated methanol formation on the Cu(111) surface via CO hydrogenation since gas-phase CO was detected for Cu-ZrO$_x$ and Cu-ZnZrO$_x$ systems (Supplementary Fig. 28). Methanol formation is favored over CO production on m-ZrO$_2$(−111), ZnZrO$_x$, and Cu$_2$-Zn$_5$ZrO$_x$ (Supplementary Fig. 29-30), in agreement with experimental observations (Fig. 3). In contrast, the energy profiles computed on Cu(111) indicate that CO$_2$ hydrogenation to methanol competes with CO formation. Still, CO desorption is favored over being further hydrogenated compatible with the lower $S_{MeOH}$ of Cu-ZrO$_x$. Moreover, DFT simulations indicate that the relative stability between HCOO* and CH$_3$O* is favored on Cu$_2$ZnZrO$_x$ over m-ZrO$_2$(−111) and ZnZrO$_x$ (Supplementary Table 6). This is in line with the rapid hydrogenation of HCOO* into methanol observed on Cu-ZnZrO$_x$ compared to Cu-ZrO$_x$ and ZnZrO$_x$ (Fig. 8g–i). Overall, based on DFT and DRIFTS, we attribute the high methanol productivity exhibited by Cu-ZnZrO$_x$ to the formation of Zn-rich low-nuclearity CuZn clusters intermixed in the ZrO$_2$ matrix. Specifically, this catalytic ensemble, which is likely stabilized by oxygen vacancies formed upon reaction, efficiently combines both oxidic (ZnO$_x$) and metallic (Cu) properties

required to effectively activate CO$_2$ and H$_2$ in nearby sites and enable concomitant and fast hydrogenation of key intermediates into methanol (Fig. 9). Finally, given the key role played by copper in promoting ZnZrO$_x$, further optimization of its content in the ternary Cu-ZnZrO$_x$ systems should be undertaken in future work.

In summary, copper is uncovered as the most effective promoter for flame-made ZnZrO$_x$ catalysts for methanol synthesis via CO$_2$ hydrogenation, rivaling state-of-the-art palladium promotion, outperforming various hydrogenation metals (i.e., 0.5 mol% Re, Co, Au, Ni, Rh, Ag, Ir, Ru, or Pt), and leading to a 2-fold increase in methanol productivity compared to unpromoted ZnZrO$_x$. In addition, the Cu-ZnZrO$_x$ catalyst exhibits a significantly more stable and superior performance over 100 h on stream compared to Cu-ZrO$_x$, where copper agglomerates into large nanoparticles, leading to inferior reactivity and methanol selectivity. In contrast, copper forms low-nuclearity metallic species on Cu-ZnZrO$_x$ upon reaction that strongly interact with zinc ensembles intermixed in the surface of the ZrO$_2$ carrier, as evidenced by combining operando XAS analysis and DFT simulations. Operando EPR spectroscopy shows that oxygen vacancy generation is particularly augmented upon reaction and closely associated with the formation of zinc-rich CuZn clusters. The resulting active catalytic ensemble comprising CuZn clusters and oxygen vacancies in the vicinity of Zn atoms efficiently integrate acid-base and H$_2$ splitting functions that enable CO$_2$ activation and barrierless homolytic H$_2$ dissociation over spatially resolved sites. In fact, this unique geometric configuration greatly fosters methanol formation through the formate (HCOO*) path, as it promotes fast hydrogenation of HCOO* to methanol, while preventing undesired long-range transport of active species and CO formation. This work sets an important milestone in the design of metal-promoted ZnZrO$_x$ catalysts and other systems based on reducible oxides, uncovering the promotional effect of a highly effective architecture based on copper, an earth-abundant element, for CO$_2$ hydrogenation to methanol. It also highlights the

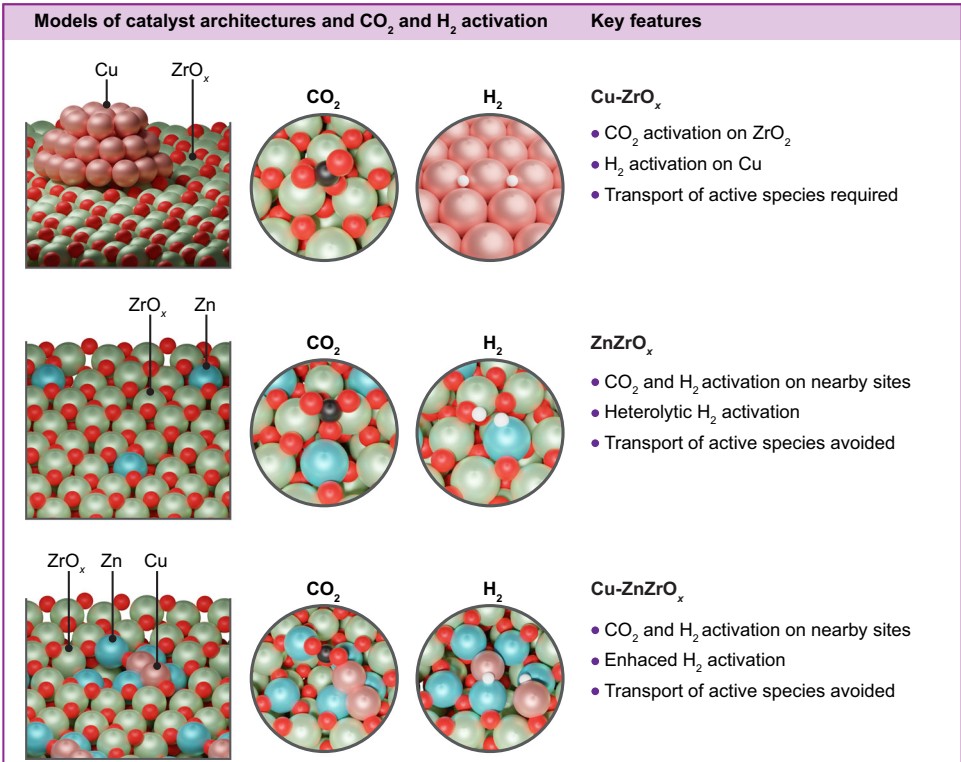

**Fig. 9 | Comparison of reference and Cu-ZnZrO$_x$ catalysts.** Summary of key geometric and reactivity features dictating the performance of Cu-ZnZrO$_x$ catalysts. Snapshots of the adsorption of CO$_2$ and the product of dissociative H$_2$ adsorption on different surfaces are shown for relevant systems. Color code of DFT models: Zn (blue), Zr (green), Cu (light pink), O (red), H (white), and C (dark gray).

potential of stable low-nuclearity metal clusters as a promising class of heterogeneous catalytic materials for diverse related applications.

# Methods

## Catalyst synthesis

$M$-ZnZrO$_x$ ($M$ = Ag, Au, Co, Cu, Ir, Ni, Pd, Pt, Re, Rh, Ru) catalysts with a nominal metal promoter and zinc contents of 0.5 and 5 mol%, respectively, were prepared by FSP. Briefly, a precursor solution of dissolved zinc, zirconium, and promoter complexes (see Supplementary Table 1 for a full list) in the desired ratio was pumped through a 0.4 mm nozzle at a flow rate of 5.0 cm$^3$ min$^{-1}$ and dispersed into a fine spray by flowing oxygen at 1.5 bar at a flow rate of 5.0 L min$^{-1}$. The spray was ignited by a supporting flame generated using 2.4 L min$^{-1}$ of oxygen and 1.2 L min$^{-1}$ of methane. Such particle-generating flames have been well characterized and reported to reach average flame temperatures of 2500–3000 K, with very fast cooling rates (ca. 10$^6$ K s$^{-1}$)[34–36]. The resulting nanoparticles were collected on a glass fiber filter (GF/A-6) and used in CO$_2$ hydrogenation without further treatment.

## Catalyst characterization

Inductively coupled plasma optical emission spectroscopy (ICP-OES) was carried out using a Horiba Ultra 2 instrument equipped with a photomultiplier tube detector. Samples were digested with the aid of microwave irradiation using a mixture of HCl (Alfa Aesar, 36 wt%), H$_2$SO$_4$ (Alfa Aesar, 95 wt%), and HF (Sigma Aldrich, 48 wt%) with a volume ratio of 2:1:0.5, followed by neutralization with a saturated solution of HBO$_3$ (Fluka, 99%). Nitrogen sorption at 77 K was carried out using a Micromeritics TriStar II analyzer. Prior to the measurements, samples were degassed at 473 K under vacuum for 12 h. The specific surface area ($S_{BET}$) was determined using the Brunauer–Emmett–Teller (BET) method. X-ray diffraction (XRD) was conducted using a Rigaku SmartLab diffractometer with a D/teX Ultra 250 detector using Cu K$\alpha$ radiation ($\lambda$ = 0.1541 nm) and operating in the Bragg-Brentano geometry. Data was acquired in the 10°–70° 2$\theta$ range with an angular step size of 0.025° and a counting time of 1.5 s per step. Temperature-programmed reduction with hydrogen (H$_2$-TPR) were measured at ambient pressure using a Micromeritics AutoChem HP II analyzer. The effluent stream of the instrument was measured using thermo-conductivity detector. High-angle annular dark-field scanning transmission electron microscopy (HAADF-STEM) and energy-dispersive X-ray (EDX) spectroscopy was carried out on a Talos F200X equipped with a Super-X EDS system and high-resolution STEM imaging on a probe-corrected HD2700CS (Hitachi), both operated at 200 kV. Catalyst powder was dispersed in ethanol, ground in a mortar, and dispersed on a copper or nickel STEM-grid covered with a lacey carbon support film. After evaporation of ethanol at room temperature, the specimen was mounted on a single-tilt holder and inserted into the microscope. Selected samples were also analyzed by HAADF-STEM-EDX using a probe-corrected Titan Themis operated at 300 kV. Operando X-ray absorption spectroscopy (XAS) experiments were conducted at the Swiss-Norwegian beamlines (SNBL, BM31) of the European Synchrotron Radiation Facility (ESRF)[37]. The X-ray beam was collimated using a double-crystal liquid nitrogen cooled Si(111) monochromator[37]. Metallic Cu and Zn foils were measured for the absolute energy calibration, using N$_2$-filled ionization chambers for optimal absorption levels. Fluorescence geometry configuration was employed for the analysis under operando conditions, using a one-element Si SDD detector with Peltier cooling. Continuous scanning was performed for both Cu (between 8880 and 9550 eV) and Zn edges (between 9550 and 10650 eV), and the step size was set to 0.5 eV, with a scan duration of 150 s. The beam size was set to 3 × 0.2 mm (horizontally × vertically). The catalyst ($m_{cat}$ = 0.013 g) was placed between two plugs of quartz wool in a quartz capillary reactor cell (1 mm outer diameter, 0.01 mm wall thickness). XAS experiments were carried out in the followings steps: (i) heating to 573 K in He ($F_T$ = 15 cm$^3$ min$^{-1}$, heating rate = 5 K min$^{-1}$, $T$ = 573 K, $P$ = 1.5 MPa, and dwell time = 30 min), (ii) switching to CO$_2$ hydrogenation conditions ($F_T$ = 15 cm$^3$ min$^{-1}$, $T$ = 573 K, $P$ = 1.5 MPa, H$_2$/CO$_2$ = 4, dwell time = 300 min), (iii) switching to He ($F_T$ = 15 cm$^3$ min$^{-1}$, heating rate = 5 K min$^{-1}$, $T$ = 573 K, $P$ = 1.5 MPa, and dwell time = 30 min) and (iv) cooling down to room temperature in He ($F_T$ = 15 cm$^3$ min$^{-1}$, heating rate = 5 K min$^{-1}$, $T$ = 303 K, $P$ = 1.5 MPa, and dwell time = 30 min). Extended X-ray absorption fine structure (EXAFS) measurements were performed at each steady state, including the fresh and used sample, as well as under CO$_2$ hydrogenation reaction conditions. Under all other conditions, including temperature ramping (steps i and iv) and after switching to different conditions (steps ii and iv, as described above), X-ray absorption near-edge structure (XANES) measurements were instead preferred, to capture the dynamic changes of the systems. The high-purity gases were dosed by a set of Bronckhorst digital mass flow controllers and the outcome was monitored on-line via a Pfeiffer Vacuum OmniStar GSD 320 O mass spectrometer with Quadera software. The resulting spectra were energy calibrated, background corrected, and normalized using the Athena program from the Demeter software suite[38]. $k^3$-weighted EXAFS spectra were fitted in the optimal $k$- and $R$-windows (Supplementary Table 5) using the Artemis program. An amplitude reduction factor ($S_0^2$) of 0.81 and 0.75 were determined by fitting of the EXAFS spectrum of a Cu and Zn foil, respectively. The scattering paths for the fitting were produced using known crystallographic structures of metallic Cu, and Zn, Cu$_5$Zn$_8$, and ZnO. Additional quasi in situ XAS measurements were performed for Re, Ir, and Pt $L_3$ edge, Au $L_2$ edge, and Co, Ni, Rh, Ag, Ru, Pd, and Cu $K$ edge. Samples were prepared under the exclusion of air by transferring catalysts samples after catalytic tests (20 h under standard conditions, see below) into quartz glass capillaries (1 mm outer diameter, 0.01 mm wall thickness, 80 mm length, Hilgenberg GmbH) and sealing of the capillaries in an inert atmosphere. The samples were placed on a stage in place of the setup used for the operando measurements and scanned in fluorescence mode the same Si SDD detector described above. In situ and operando electron paramagnetic resonance (EPR) spectroscopy experiments were performed using a custom-built setup (microwave frequency = 9.2 GHz, center field = 300 mT, sweep width = 570 mT, modulation amplitude = 3 G, modulation frequency = 100 kHz, microwave power = 1.986 mW, power attenuation = 20 dB, conversion time = 86.55 ms, time constant = 20.48 ms). The catalyst sample ($m_{cat}$ = 50 mg) was loaded into a quartz capillary ($d_i$ = 0.8 mm) and placed inside an EPR quartz tube (Wilmad; $d_i$ = 2.8 mm). The EPR tube was housed at the center of a homemade water-cooled high-temperature resonator[39], which was installed into a continuous wave (CW) EPR spectrometer (Bruker EMX) operating at X-band frequencies. The reactor was pressurized to 1 MPa, heated in a He flow (20 cm$^3$ min$^{-1}$) to the desired temperature ($T$ = 573 K) and kept at this condition for 2 h, while continuously recording EPR spectra. It was subsequently cooled down to room temperature and an EPR spectrum was recorded. The reactor was then heated up again to 573 K and left to stabilize for 30 min. The reaction mixture (H$_2$/CO$_2$ = 4) was then admitted and was kept flowing (20 cm$^3$ min$^{-1}$) for 2 h. All gases were dosed by a set of digital mass flow controllers and the reactor outlet was monitored online using a Pfeiffer Vacuum Thermo-Star GSD 320 T1 mass spectrometer. The EPR spectra were continuously acquired upon flowing the gases and separately stored, using a 2D acquisition mode, thus enabling a time-resolved monitoring of the process. The reactor was eventually cooled down and EPR spectra were recorded at room temperature. He and air were then subsequently admitted and the EPR spectrum was monitored until no changes were detected. Operando diffuse reflectance infrared Fourier transform spectroscopy (DRIFTS) was conducted in a custom-built setup comprising a gas dosing unit, back-pressure regulators, a Pfeiffer Vacuum OmniStar GSD 350 O$_2$ mass spectrometer for online product analysis, and a Bruker INVENIO-S spectrometer equipped with a liquid nitrogen cooled mercury cadmium telluride (MCT) detector. In a typical experiment, 50 mg of the undiluted sample was packed into a high-

temperature reaction chamber with ZnSe windows (Harrick Scientific), which was mounted in a Praying Mantis diffuse reflection attachment (Harrick Scientific). After purging the sample with flowing He (20 cm$^3$ min$^{-1}$, PanGas, 4.6) for 60 min, the reaction chamber was pressurized to 1.5 MPa and heated to 573 K with a ramp of 5 K min$^{-1}$. Subsequently, spectra were continuously acquired at a spectral resolution of 2 cm$^{-1}$ by accumulating 128 scans every 30 s and were normalized against a KBr background taken at 573 K and 1.5 MPa. The gas feed was then switched to a mixture of H$_2$ (PanGas, 5.0) and CO$_2$ (40 vol% in H$_2$, Messer, 4.5) with a molar H$_2$/CO$_2$ ratio of 4 at a total flow rate of 20 cm$^3$ min$^{-1}$. After 90 min under reaction conditions, the CO$_2$ flow was stopped and H$_2$ (16 cm$^3$ min$^{-1}$) balanced with He (4 cm$^3$ min$^{-1}$) was fed into the reaction chamber for an additional 90 min. The raw data obtained with Bruker OPUS 8.2 software were processed and difference spectra as well as intensity profiles were generated with a self-coded Python routine, available as open-source on GitHub (https://github.com/philpreikschas/operando-ir).

## Catalyst evaluation

The gas-phase hydrogenation of CO$_2$ to methanol was conducted using a PID Eng&Tech high-pressure continuous-flow setup comprising four parallel fixed-bed reactors, as described elsewhere[40]. Undiluted catalysts (mass, $m_{cat}$ = 0.1 g; particle size = 0.2-0.4 mm) were loaded into each reactor tube (internal diameter 4 mm), held in place by a quartz-wool bed set on a quartz frit, and purged in flowing He (40 cm$^3$ min$^{-1}$, PanGas, 4.6) for 30 min. Under the same flow, the pressure was increased to 5 MPa for a leak test, which was followed by heating up the catalyst bed to 573 K (5 K min$^{-1}$). The reaction was carried out by feeding a mixture of H$_2$ (PanGas, 5.0), CO$_2$ (40 vol% in H$_2$, Messer, 4.5), with a molar H$_2$/CO$_2$ ratio of 4 at 573 K, 5 MPa, and a gas hourly space velocity (GHSV) of 24,000 cm$^3$ h$^{-1}$ g$_{cat}^{-1}$, unless stated otherwise. The selectivity of the catalysts was compared at a constant degree of CO$_2$ conversion ($X_{CO_2}$) as described in Fig. 3b by adjusting the GHSV for each system.

The effluent streams were analyzed by gas chromatography every 1 h. Response factors ($F_i$) for each compound $i$, respective to the internal standard (20 vol% C$_2$H$_6$ in He, Messer, purity 3.5), in the GC analysis were determined by Eq. (1):

$$F_i = \frac{A_{C_2H_6} / n_{C_2H_6}^{in}}{A_i / n_{in}^{in}} \qquad (1)$$

where $A_i$ is the integrated area determined for the peak of compound $i$ and $n_{in}^{in}$ is the corresponding known molar flow at the reactor inlet. An average of 5 points around the expected analyte concentration was used. The unknown effluent molar flow of compound $i$ ($n_i^{out}$) was determined using Eq. (2):

$$n_i^{out} = \frac{A_i \times F_i}{A_{C_2H_6}} \times n_{C_2H_6}^{out} \qquad (2)$$

where $n_{C_2H_6}^{out}$ is the known flow of C$_2$H$_6$ at the reactor outlet.

Conversion ($X_i$), selectivity ($S_i$), and methanol production rate ($r_{MeOH}$) were calculated using Eqs. (3–5):

$$X_i = \frac{n_i^{in} - n_i^{out}}{n_i^{in}} \qquad (3)$$

$$S_i = \frac{n_i^{out}}{n_{CO_2}^{in} - n_{CO_2}^{out}} \qquad (4)$$

$$r_{MeOH} = \frac{n_{MeOH}^{out}}{m_{cat}} \qquad (5)$$

The methanol space-time yield (STY) is the product of $r_{MeOH}$ and the molar weight ($M_W$) of methanol (32.04 g mol$^{-1}$) as stated in Eq. (6):

$$STY = r_{MeOH} \times M_W \qquad (6)$$

The carbon balance was determined for each experiment according to Eq. (7):

$$E_C = \left(1 - \frac{n_{CO_2}^{out} + n_{MeOH}^{out} + n_{CO}^{out}}{n_{CO_2}^{in}}\right) \qquad (7)$$

and was always within a 5% margin.

## Computational details

Density function theory (DFT) simulations were conducted with Vienna ab initio simulation package (VASP)[41] and the Perdew-Burke-Ernzerhof (PBE) density functional[42,43]. Valence electrons were expanded from a plane-wave basis set with a kinetic cut-off energy of 500 eV while core electrons were described with projector augmented-wave (PAW) pseudopotentials[44,45]. The Brillouin zone was sampled with a Γ-centered mesh with a reciprocal grid size narrower than 0.037 Å$^{-1}$, which was obtained with the Monkhorst-Pack method[46]. Lattice parameters for $m$-ZrO$_2$, Cu, Zn, CuO, ZnO, and Cu$_5$Zn$_8$ were optimized with a kinetic energy cut-off of at least 700 eV. Then, slab models for $m$-ZrO$_2$(−111)[4,47,48], Cu(111), and Cu$_5$Zn$_8$ (110)[49,50] terminations were cleaved from the optimized bulks and represented with $p(1 \times 1)/p(2 \times 2)$, $p(4 \times 4)$, and with $p(1 \times 1)$ expansions, respectively. In all cases, the two bottommost layers were kept fixed and the two outermost were allowed to relax. We added a vacuum region of 15 Å between slabs and a dipole correction along the $z$ axes was applied to account for the asymmetry in the relaxations[51]. The incorporation of Zn and Cu, as well as the formation of oxygen vacancies were assessed at different sites of $m$-ZrO$_2$(−111) following the procedure employed in our previous work[4]. Additionally, the deposition of single atoms and small clusters (up to 4 atoms) of Cu was explored in models of pure and Zn-doped $m$-ZrO$_2$ surfaces. The adsorption energies of key molecules (CO$_2$, CO, H$_2$, HCOO*, and CH$_3$O*) was carried out in relevant surfaces to rationalize the catalytic properties of ZnZrO$_x$, Cu-ZrO$_x$, and Cu-ZnZrO$_x$. Clean surfaces and gas-phase CO$_2$, H$_2$, and CO molecules were employed as thermodynamic sinks. The adsorptions of HCOO* and CH$_3$O* were performed with the adsorption of an H atom in nearby sites as OH or $M$H to avoid the adsorption of fragments with unpaired electrons that modify the electronic properties of semiconducting metal-oxides[52]. Furthermore, the effect of weak long-range interactions and the self-interaction error on the adsorption energies were evaluated (Supplementary Tables 8 and 9)[53]. Dispersion corrections were included via Grimme's D3 approach while a Hubbard correction[54] by means of the Dudarev approach[55] with a U$_{eff}$ value of 4.00 eV was applied on the $d$-states of Zr atoms[56–59].

Energy profiles for CO$_2$ hydrogenation to methanol and the competitive RWGS to form CO were computed on selected models of the ZnZrO$_x$, Cu-ZrO$_x$, and Cu-ZnZrO$_x$ catalysts. Clean surfaces and gas-phase CO$_2$, H$_2$, and H$_2$O molecules were used as thermodynamic sinks. The climbing image nudged elastic band (CI-NEB)[60] was employed to locate transition states. Their nature was confirmed by computing numerical frequencies with a step size of ± 0.015 Å.

## Data availability

Data presented in the main figures of the manuscript are publicly available through the Zenodo repository (entry: 8309938, link: https://doi.org/10.5281/zenodo.8309938)[61]. Inputs and outputs for all DFT simulations can be found online in the ioChem-BD repository[62,63], at https://iochem-bd.iciq.es/browse/review-collection/100/64621/fd52044500da8d298dc081fd. Further data supporting the findings of this study are available in the Supplementary Information and source

data. All other relevant source data are available from the corresponding author upon request. Source data are provided with this paper.

## Code availability

Source data of the Python application used for processing and analyzing operando infrared spectroscopy data are open-sourced on GitHub (https://github.com/philpreikschas/operando-ir) and additionally available in the Zenodo repository (https://doi.org/10.5281/zenodo.10818472)[64]

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

## Acknowledgements

This publication was created as part of NCCR Catalysis (grant number 180544), a National Centre of Competence in Research funded by the Swiss National Science Foundation. The Scientific Center for Optical and Electron Microscopy (ScopeM) at ETH Zurich is thanked for access to their facilities. Mr. Dario Faust Akl and Mr. Henrik Eliasson are thanked for acquiring the HAADF-STEM-EDX data. T.Z. thanks the Agency for Science, Technology and Research (A*STAR) Singapore for support through a graduate fellowship. The Spanish Ministry of Science and Innovation is acknowledged for financial support (PRE2019-088791, PID2021-122516OB-I00, and Severo Ochoa Grant MCIN/AEI/10.13039/501100011033 CEX2019-000925-S) and the Barcelona Supercomputing Center-MareNostrum (BSC-RES) for providing generous computer resources. Pol Sanz Berman is thanked for his support with graphic design. The Swiss Norwegian beamlines (SNBL, ESRF) are acknowledged for the provision of beamtime and its staff for invaluable support.

## Author contributions

T.P.A. and J.P.-R. conceived and coordinated all stages of this research. T.P.A., G.G., J.M.V, N.L., S.M., and J.P.-R. wrote the article with input from all other co-authors. T.P.A. and Z.R.B. conducted catalyst characterization and evaluation. J.M.V and N.L. conducted the density functional theory simulations. F.K. conducted the electron microscopy analyses. M.A. and G.J. conducted the electron paramagnetic resonance spectroscopy studies. G.G. evaluated X-ray absorption spectroscopy data. P.O.W., R.N.G., and W.J.S. prepared the catalysts using the flame spray pyrolysis method. T.Z. conducted kinetic analyses. P.P. supervised acquisition and evaluated diffuse reflectance infrared Fourier transform spectroscopy data. J.P.-R. supervised the entire project and managed resources and funding. All the authors provided inputs to the manuscript and approved the final version.

## Competing interests

The authors declare no competing interests.
