## [Peer Review File · Nature Communications]

Low-nuclearity CuZn ensembles on ZnZrOx catalyze methanol synthesis from CO₂REVIEWER COMMENTS

Reviewer #1 (Remarks to the Author):

This work investigate promotion of flame-made ZnZrOx catalysts by relevant hydrogenation metals (0.5 mol% Re, Co, Au, Ni, Rh, Ag, Ir, Ru, Pt, Pd, and Cu) for CO₂ hydrogenation to methanol. Detailed microscopy, kinetic, stability, operando spectroscopy analyses, and theoretical simulations are applied to gather understanding of copper promotion. Some interesting and important results were reported and discussed. However, the construction of the ZrO₂ model in theoretical calculations is not appropriate. In addition, some conclusions have not received strong support and are not convincing. Therefore, there are some concerns that the authors should address well before further considering publication. The detailed comments are as follows:

1. Please list and compare the deactivation rates of Cu-ZnZrOx and Cu-ZrOx catalysts. Within TOS of 60 hours, the catalytic performance of Cu-ZnZrOx has been continuously decreasing. Therefore, it is difficult to conclude that Cu-ZnZrOx is more stable than Cu-ZrOx.
2. The structure of ZnZrOx is unstable in the reaction. The t-ZrO₂ was partially transforms into the monoclinic structure upon reaction. However, this phenomenon does not appear to have been observed in other reported solid solution ZnZrOx oxides. Is the stability of oxides related to the preparation method? If so, is this a disadvantage of the FSP preparation method compared to the co precipitation method?
3. The XRD peak intensity changed with different spent catalysts. Do different metal additives have an impact on the ratio of t-ZrO₂/m-ZrO₂ on activated catalyst? In addition, does it affect the catalytic performance for CO₂ hydrogenation to methanol?
4. Is there any similar literature support for the conclusion that pure metallic or Zn-containing phases can be formed for Pd-ZnZrOx catalyst drawn from the H₂-TPR results?
5. For ZnZrOx, more oxygen vacancies on the surface of catalysts can be obtained after reduction in hydrogen. I suggest the author explore the effect of reduction temperature on reaction behavior.
6. The author mentioned that exposure to the reaction mixture (CO₂+H₂) led to a decrease in the Zr³⁺ signal. However, no change can be observed during the initial stage of CO₂ hydrogenation reaction in Fig. 3a. Why?
7. The t-ZrO₂ was partially transforms into the monoclinic structure upon reaction. This means that both t-ZrO₂ and m-ZrO₂ phases were present in the spent catalyst. However, the author chose m-ZrO₂ model in DFT calculations to study the reaction mechanism, which is unreasonable and meaningless.

Reviewer #2 (Remarks to the Author):

Extensive experimental work, along with DFT calculations, was conducted to comprehend the promising catalytic performance of CuZnZrO₂ for methanol synthesis from CO₂. This reaction offers the potential to convert waste CO₂ into the value-added product methanol, usable either directly or as a precursor to other valuable organic compounds. However, for this process to be viable on the

industrial scale, an affordable, active, and selective catalyst is crucial. The current work presents a significant step towards that direction.

The experimental work demonstrates potential for publication in Nature Communications. However, the computational aspects exhibit several shortcomings that need to be addressed satisfactorily before considering the work for publication in Nature Communications as a whole.

1) Several atomistic models are presented to simulate experimental catalysts; however their justification remains largely insufficient. The selection of a five Zn-atom cluster embedded on ZrO₂ lacks clarity in its rationale: why favor a five-atom cluster over three or six atom Zn cluster in calculations? The question arises whether these Zn atoms ideally form close contacts or if five isolated Zn-O vacancy pairs might be more stable computationally? Additionally, based on Figure 5 the formation of a five-atom Zn cluster is accompanied by the five oxygen vacancies very close to each other. How the formation of five oxygen vacancies is justified? Would it be possible to form that cluster with only four or three vacancies. How stable the vacancy structure is under methanol synthesis, which generates OH groups capable of filling these vacancies?

Utilization of the favorable formation energy of the bulk alloy Cu₅Zn₈ as an indication of Cu-Zn affinity raises uncertainties. The rationale behind selecting the Cu:Zn ratio 5:8 for bulk and 2:5 for ZrO₂ embedded structure despite 1:5 Cu:Zn ratio is energetically unfavorable for the embedded structure, remains unclear. Furthermore, Figure S14 exhibits various Cu-Zn₅-ZrO₂ models; however, the computational approach employed to explore these structures for a comprehensive screening of structural possibilities is unspecified.

Finally, the Supplementary material, indicates (page 3) that the formation of larger clusters is unfavorable. Yet, based on the energies depicted in Figure S14, even the incorporation of the first Cu atom into Zn₅-ZrO₂ structure appears to energetically unfavorable. Overall, clarity regarding these structural models and their outcomes is necessary for comprehensive understanding.

2) Figure 7 provides a summary of the adsorption energies of key experimentally identified intermediates. I suspect that the adsorption energies refer to the energetically most favorable energies although this is not said in the figure caption. It would be beneficial to include the corresponding adsorption structures for CO, HCOO and CH₃O structures at least in Supplementary material.

There are notable concerns regarding these adsorption energies. Firstly, the computational approach lacks the consideration of VdW corrections, neglecting weak long-range interactions between the adsorbate and the surface. This omission can potentially introduce errors into calculated adsorption energies. Second, the cooperative effects present for adsorption on oxides are unsystematically considered in the present work, please see J. Phys. Chem. C 121, 18608 (2017). According to the computational details, the adsorption of HCOO and H₃CO is computed in the presence of surface H, while no surface-H is present for CO₂ and CO adsorption. This discrepancy considering surface-H presence or absence can lead to significant differences in adsorption energies of carbon-based species. Consequently, in this study, the computed adsorption energies might not directly comparable due to varying conditions.

It is important to note that the presence or absence of terminal OH groups formed during the reaction could also influence the acid-base balance and subsequent adsorption energies. These factors collectively warrant attention for a more comprehensive evaluation of the adsorption phenomena.

Furthermore, the choice of excluding Hubbard U correction from the calculations must be computationally justified.

3) Calculations have exclusively focused on thermodynamic aspects of methanol synthesis. I warmly recommend extending these computations to include the determination of activation energies, specifically encompassing H₂ dissociation as well as the hydrogenation of HCOO and H₃CO. This

broader computational scope will not only enhance the validation of catalyst models, but also provide additional support for the experimental findings.

Minor issues:

1) When discussing heterolytic H₂ dissociation, Figure 9 should be cited not Figure 7.

Reviewer #3 (Remarks to the Author):

The manuscript reported a metal promotion in ZnZrOx catalyst used for CO₂ hydrogenation to methanol. It was found that Cu had the best effective. The authors suggested that the low-nuclearity CuZnx clusters on the ZrO₂ surface are responsible for the effect.

The manuscript could be published after addressing the comments

1. The promoted Cu concentration should be optimized in this work.
2. CuZnOx catalyst should also be placed to the comparison object.
3. Is there additional evidence to support or affirm the Cu structure in the Cu-ZnZrOx catalyst. Specifically, single atom, di-atom, or cluster? By contrast, Pd seems to in the forms of particale?
4. It can be seen that Cu-ZnZrOx catalyst exhibits less stability than ZnZrOx catalyst?
5. Weather the Cu promotion changed the pathway from CO₂ to methanol? HCOO* pathway or CO* pathway?

Response to Reviewers for Manuscript NCOMMS-23-45274-T

Code: Comments in *blue* | Replies in black | Actions in **bold** | Revision location **highlighted**

Note: Indicated page, line, figure, or reference numbers refer to the revised manuscript and/or supplementary information with changes highlighted.

Reviewer #1

This work investigate promotion of flame-made ZnZrO_x catalysts by relevant hydrogenation metals (0.5 mol% Re, Co, Au, Ni, Rh, Ag, Ir, Ru, Pt, Pd, and Cu) for CO₂ hydrogenation to methanol. Detailed microscopy, kinetic, stability, operando spectroscopy analyses, and theoretical simulations are applied to gather understanding of copper promotion. Some interesting and important results were reported and discussed. However, the construction of the ZrO₂ model in theoretical calculations is not appropriate. In addition, some conclusions have not received strong support and are not convincing. Therefore, there are some concerns that the authors should address well before further considering publication. The detailed comments are as follows:

We warmly thank the Reviewer for recognizing the relevance and quality of our study. Their valuable suggestions helped us further improve the manuscript, as detailed below.

1. Please list and compare the deactivation rates of Cu-ZnZrO_x and Cu-ZrO_x catalysts. Within TOS of 60 hours, the catalytic performance of Cu-ZnZrO_x has been continuously decreasing. Therefore, it is difficult to conclude that Cu-ZnZrO_x is more stable than Cu-ZrO_x.

The observed induction period for the Cu-ZnZrO_x catalyst, characterized by a gradual drop in activity with the first 60 h of operation, is a common phenomenon due to the restructuring and equilibration of the catalyst at elevated temperature and pressure. A similar behavior is observed for the Cu-ZrO_x catalyst, albeit requiring a shorter time (ca. 20 h), likely due to the simpler architecture of the latter catalyst. It is important to note that beyond the induction period, Cu-ZnZrO_x demonstrates stable performance, greatly surpassing that of the Cu-ZrO_x system. **We have included this remark in the revised manuscript (page 7, lines 4-9).**

2. The structure of ZnZrO_x is unstable in the reaction. The t-ZrO₂ was partially transforms into the monoclinic structure upon reaction. However, this phenomenon does not appear to have been observed in other reported solid solution ZnZrO_x oxides. Is the stability of oxides related to the preparation method? If so, is this a disadvantage of the FSP preparation method compared to the co precipitation method?

We thank the Reviewer for raising this relevant point. The partial transformation of tetragonal(*t*) ZrO₂ into monoclinic(*m*) ZrO₂ is a well-documented phenomenon for zirconia-containing catalytic materials prepared by the flame-spray pyrolysis (FSP) method used in our study (see *Nat. Commun.* **2022**, 13, 5610; *Adv. Energy Mater.* **2023**, 13, 2204122). Differently, ZnZrO_x systems reported in the literature are often prepared by coprecipitation, which leads to incorporation of zinc into the bulk structure of ZrO₂, forming a solid solution that stabilizes the tetragonal phase and thus prevents its transformation into the monoclinic form. Similar behavior is also known for a wide range of cations (*i.e.*, Y³⁺, Si⁴⁺, Ca²⁺) forming solid solutions with zirconia (see K.-O. Axelsson *et al.*, *Appl. Surf. Sci.* **1986**, 25, 217-230, A. Bernasik *et al.*, *J. Phys. Chem. Solids* **2002**, 63, 233-239, M. Asri Idris *et al.*, *J. Phys. Chem. C* **2012**, 116, 10950–10958). In contrast, FSP tends to maximize zinc location and dispersion at the outmost layers of ZrO₂, leading to a less stabilized *t*-phase and thus enabling the thermodynamically favored *t*-to-*m* transformation to occur (see *Adv. Energy Mater.* **2023**, 13, 2204122). As shown in that study, this distinctive feature of flame-made catalysts compared to coprecipitated renders a catalytic advantage. In fact, FSP produces ZnZrO_x catalysts with superior performance compared to their CP counterparts,

owing to their higher surface areas coupled to improved zinc utilization and maximized location and atomic dispersion at surface lattice positions of the ZrO₂ surface phase. **We further stressed this point in the revised manuscript (page 5, lines 12-20).**

*3. The XRD peak intensity changed with different spent catalysts. Do different metal additives have an impact on the ratio of *t*-ZrO₂/*m*-ZrO₂ on activated catalyst? In addition, does it affect the catalytic performance for CO₂ hydrogenation to methanol?*

Metal content rather than its identity has been reported to strongly influence the *t*-ZrO₂/*m*-ZrO₂ ratio in flame-made as well as impregnated zirconia-containing catalysts (see C. Yang *et al.*, *J. Am. Chem. Soc.* **2020**, 142, 19523; *Nat. Commun.* **2022**, 13, 5610; *Adv. Energy Mater.* **2023**, 13, 2204122). Although XRD analysis often evidenced the presence of both *m*- and *t*-ZrO₂, UV Raman spectroscopy, which probes the zirconia structure confined at the outmost surface layers, revealed that the surface of *t*-ZrO₂ particles undergoes a transformation into the *monoclinic* phase. This was particularly evident in the used 5ZnZrO_x FSP catalyst (see *Adv. Energy Mater.* **2023**, 13, 2204122), which served as the optimal base formulation to assess promotion by different metals in the present study. Due to the lower metal promoter content (0.5 mol%) added to the 5ZnZrO_x base material, it is very likely that the surface of all metal-promoted 5ZnZrO_x catalysts predominantly contains *m*-ZrO₂. Therefore, the *t*-ZrO₂/*m*-ZrO₂ ratio is not expected to play any role on the catalytic performance for CO₂ hydrogenation. **We have included this new discussion and corresponding references in the supplementary information (page 4, lines 9-20).**

4. Is there any similar literature support for the conclusion that pure metallic or Zn-containing phases can be formed for Pd-ZnZrO_x catalyst drawn from the H₂-TPR results?

Thank you for drawing our attention to this point. Several reports supporting the formation of pure metallic palladium or Zn-containing phases in Pd-ZnZrO_x and Pd-ZnO catalysts under CO₂ hydrogenation conditions are available in the literature (see C. Huang *et al.*, *ACS Appl. Energy Mater.* **2021**, 4, 9258–9266; M. Zabilskiy *et al.*, *Angew. Chem. Int. Ed.* **2021**, 60, 17053–17059; K. Lee *et al.*, *Appl. Catal. B: Environ* **2022**, 304, 120994;). Additionally, as mentioned in the original manuscript, Pd-Zn contributions can be reliably assigned in the extended X-ray absorption fine structure spectra (EXAFS, Pd K-edge, **Supplementary Fig. 10**) of Pd-ZnZrO_x, confirming the formation of PdZn_x alloy and thus corroborating H₂-TPR results. **We have included the additional references (page 6, line 12) and further clarified these points in the revised manuscript (page 7, lines 14-17).**

5. For ZnZrO_x, more oxygen vacancies on the surface of catalysts can be obtained after reduction in hydrogen. I suggest the author explore the effect of reduction temperature on reaction behavior.

Following the Reviewer's suggestion, key catalytic systems were first pretreated in H₂ at two distinct temperatures for 1 h (*i.e.*, 573 and 623 K) followed by assessing their performance in CO₂+H₂ mixtures at 573 K and 5 MPa. No differences in methanol productivity were observed among systems evaluated with or without prior reduction in H₂ (**Supplementary Fig. X**), indicating that such pretreatment step likely does not induce any relevant change in the formation of oxygen vacancies. While other temperatures could be explored, temperatures higher than 623 K will likely induce sintering of the metal promoter, leading to detrimental effects on performance. **We have added these results and corresponding discussion to the revised supplementary information (Supplementary Fig. 12) (page 28, lines 2-9).**

6. The author mentioned that exposure to the reaction mixture (CO_2+H_2) led to a decrease in the Zr^{3+} signal. However, no change can be observed during the initial stage of CO_2 hydrogenation reaction in Fig. 3a. Why?

Operando EPR experiments evidenced that the concentration of Zr^{3+} species on ZnZrO_x slowly decreased upon exposure to the CO_2+H_2 mixture (Fig. 6e). However, as observed in a previous study (see *Adv. Energy Mater.* **2023**, 13, 2204122), Zr^{3+} does not effectively impact the catalytic ensemble properties, acting as spectator species. Hence, though Zr^{3+} signal decreased upon reaction, this would not translate into a change in performance of the ZnZrO_x catalyst for CO_2 hydrogenation. **We have included this discussion in the revised manuscript (page 12, lines 20-21).**

7. The $t\text{-ZrO}_2$ was partially transforms into the monoclinic structure upon reaction. This means that both $t\text{-ZrO}_2$ and $m\text{-ZrO}_2$ phases were present in the spent catalyst. However, the author chose $m\text{-ZrO}_2$ model in DFT calculations to study the reaction mechanism, which is unreasonable and meaningless.

Indeed, XRD analysis showed that $t\text{-ZrO}_2$ partially transforms into $m\text{-ZrO}_2$ under CO_2 hydrogenation conditions in all the examined metal-promoted ZnZrO_x catalysts (Supplementary Fig. 3). However, as emphasized in the response to **comment #3**, our previous UV-Raman spectroscopic studies revealed that the surface of the ZnZrO_x catalyst (see *Adv. Energy Mater.* **2023**, 13, 2204122), which is the most relevant part of the material for catalytic purposes, fully transforms into the *monoclinic* phase upon reaction. Furthermore, DFT simulations indicated that both $t\text{-ZnZrO}_x$ and $m\text{-ZnZrO}_x$ surfaces selectively hydrogenate CO_2 to methanol, with the latter being slightly more active due to electronic and geometric effects (see *Adv. Energy Mater.* **2023**, 13, 2204122). Finally, other studies have shown that the *monoclinic* phase is thermodynamically favored under reactions (see M. Ruhle, *Adv. Mater.* **1997**, 9, 195) and is active for CO_2 hydrogenation to methanol (see K. Taek Jung *et al.*, *Catal. Lett.* **2002**, 80, 63; A. Lempelto *et al.*, *Catal. Sci. Technol.* **2023**, 13, 4387; A. Arandia *et al.*, *Appl. Catal. B: Environ.* **2023**, 321, 122046). Consequently, our modeling is based on slabs representing the $m\text{-ZrO}_2(-111)$, the most stable termination of this polymorph (see A. Christensen *et al.*, *Phys. Rev. B* **1998**, 58, 8050; C. Ricca *et al.*, *J. Comput. Chem.* **2015**, 36, 9). **We have included this discussion in the revised supplementary information (page 4, lines 9-20)** to clarify the selection of $m\text{-ZrO}_2$ phase in our DFT simulations.

Reviewer #2

Extensive experimental work, along with DFT calculations, was conducted to comprehend the promising catalytic performance of CuZnZrO₂ for methanol synthesis from CO₂. This reaction offers the potential to convert waste CO₂ into the value-added product methanol, usable either directly or as a precursor to other valuable organic compounds. However, for this process to be viable on the industrial scale, an affordable, active, and selective catalyst is crucial. The current work presents a significant step towards that direction. The experimental work demonstrates potential for publication in Nature Communications. However, the computational aspects exhibit several shortcomings that need to be addressed satisfactorily before considering the work for publication in Nature Communications as a whole.

We appreciate the Reviewer's acknowledgement of the relevance and timely character of our study. Regarding the computational aspects, we recognize the importance of addressing the representativeness of the models. To ensure compatibility with experimental observations and fulfill energetic requirements based on the DFT analysis, we have evaluated over 200 different configurations. Specific actions are detailed below.

Several atomistic models are presented to simulate experimental catalysts; however their justification remains largely insufficient. The selection of a five Zn-atom cluster embedded on ZrO₂ lacks clarity in its rationale: why favor a five-atom cluster over three or six atom Zn cluster in calculations?

We appreciate the Reviewer's attention to the atomistic models presented in our study. Given the complexity of the multicomponent Cu-ZnZrO_x system, our modeling approach was guided by in-depth experimental characterization. The characterization revealed the formation of low-nuclearity zinc-rich CuZn clusters stabilized on the ZrO₂ support, with Cu present in a metallic state and associated with partially reduced Zn atoms. We initially started with a pool of at least 100 configurations that were narrowed down to a single model based on two criteria: (i) the compatibility with the experimental observations (cluster nature); and (ii) the DFT-computed energies were reasonable (*i.e.*, processes thermodynamically favored). To ensure the robustness of our approach, **we have performed additional DFT simulations to assess the relative stability of several models containing 3 to 6 Zn atoms incorporated into the ZrO₂ lattice with oxygen vacancies, as well as the tendency of Cu atoms to bind to the Zn aggregates (Supplementary Fig. 21)**. Adsorption of a single Cu atom is particularly favored on the structure with 5 Zn and 5 oxygen vacancies incorporated into the ZrO₂ surface. Furthermore, adsorption of a second Cu atom is exothermic by 0.50 eV, resulting in a CuZn pattern. Notably, this structure is very similar to that exhibited by the one of the most stable brass in its lowest surface Cu₅Zn₈(110), demonstrating that the ensembles are robust since they are driven by strong thermodynamic preference (**Fig. 5**). Therefore, the Cu₂-Zn₅ZrO_x model was identified as the most stable among alternative models with 3, 4, and 6 Zn atoms. Since the latter fall within a range of 0.58 eV, the formation of other low-nuclearity zinc-rich clusters with catalytic properties similar to Cu₂-Zn₅ZrO_x cannot be fully ruled out. **We have included this discussion on the selection of the Cu₂-Zn₅ZrO_x model in the revised manuscript (page 9, lines 25-26 and page 10, lines 1-11) and supplementary information (page 5, lines 16-28 and page 7, lines 1-5).**

The question arises whether these Zn atoms ideally form close contacts or if five isolated Zn-O vacancy pairs might be more stable computationally?

The formation of vacancies close to Zn is favored over other positions of the *m*-ZrO₂. Regarding the Zn distribution within the *m*-ZrO₂ matrix, no thermodynamic preference was observed as long as the vacancies are close to the Zn atoms (see *Adv. Energy Mater.* **2023**, 13, 2204122). Accordingly, **we investigated 51 configurations representing the incorporation of two Zn atoms accompanied by two oxygen vacancies at different sites within the surface lattice of ZrO₂ (Supplementary Fig. 17)**. No correlation between the stability of Zn-O_{vac} pairs and its distance was identified. Moreover, Zn

incorporation into the ZrO_2 surface is contingent on the specific location of Zn and oxygen vacancies, as indicated by our previous thorough study of Zn incorporation into $m\text{-ZrO}_2(-111)$ surface with one oxygen vacancy (see *Adv. Energy Mater.* **2023**, 13, 2204122). The potential energies of the various structures were determined to fall within a range of 2.18 eV. Consequently, considering the experimental findings and the absence of a correlation between Zn- O_{vac} pairs distance and their relative stability, the $\text{Cu}_2\text{-Zn}_5\text{ZrO}_x$ model was selected to best represent the Cu-Zn ZrO_x catalyst. **We have clarified these points in the supplementary information (page 6, lines 6-14).**

Additionally, based on Figure 5 the formation of a five-atom Zn cluster is accompanied by the five oxygen vacancies very close to each other. How the formation of five oxygen vacancies is justified? Would it be possible to form that cluster with only four or three vacancies.

In *Adv. Energy Mater.* **2023**, 13, 2204122, we systematically explored Zn incorporation at all non-equivalent sites of the $m\text{-ZrO}_2(-111)$ surface with 0-2 oxygen vacancies. Zn incorporation with one oxygen vacancy was found to be favored by 2.87 (2.01) eV over the analogous structures without (with two) oxygen vacancies. Furthermore, **we have compared the relative stability of a five-atom Zn cluster with 5 oxygen vacancies with respect to alternative structures with 3 or 4 vacancies (Supplementary Fig. 18).** The model comprising 5 oxygen vacancies is favored by more than 2.8 eV over the most stable structures containing 3 and 4 oxygen vacancies, respectively. **We have included this discussion in the supplementary information (page 6, lines 18-23).**

How stable the vacancy structure is under methanol synthesis, which generates OH groups capable of filling these vacancies?

We have evaluated the tendency of OH groups to fill oxygen vacancies located at the $\text{Cu}_2\text{-Zn}_5\text{ZrO}_x$ ensemble. The Gibbs free energy (ΔG) associated with the adsorption and dissociation of a water molecule from the gas phase on $\text{Cu}_2\text{-Zn}_5\text{ZrO}_x$ at 573 K healing an oxygen vacancy on the surface and forming a nearby OH is almost thermoneutral (out of the three configurations investigated two are exergonic $\Delta G = -0.14$ and -0.32 eV and one endergonic $\Delta G = 0.21$ eV). These results suggest that some oxygen vacancies could potentially be filled by OH groups under reaction conditions, but they will be easily removed and replenish during the process. **We have emphasized this point in the supplementary information (page 6, lines 23-27) and added Supplementary Fig. 19.**

Utilization of the favorable formation energy of the bulk alloy Cu_5Zn_8 as an indication of Cu-Zn affinity raises uncertainties. The rationale behind selecting the Cu:Zn ratio 5:8 for bulk and 2:5 for ZrO_2 embedded structure despite 1:5 Cu:Zn ratio is energetically unfavorable for the embedded structure, remains unclear.

As highlighted in the original submission, the adsorption of a single Cu atom at a Zn site incorporated within the surface lattice of $m\text{-ZrO}_2$ is more favored compared to its deposition on zinc-free ZrO_2 surfaces (**Fig. 5**). This is also corroborated by the new DFT calculations on the adsorption of Cu on similar structural motifs containing 3, 4, 5, or 6 Zn atoms (**Supplementary Fig. X**). Besides the exothermic formation energy of the bulk Cu_5Zn_8 alloy ($E_{\text{Cu}_5\text{Zn}_8} = -0.26$ eV), **we have computed the formation energy of an additional alloy with a different Cu:Zn ratio (1:1)**, which is also exothermic ($E_{\text{CuZn}} = -0.17$ eV), further confirming the Cu-Zn affinity. **We have highlighted the CuZn affinity and the different computational approaches applied to determine it in the supplementary information (page 5, lines 22-28 and page 6, lines 1-5) and revised manuscript (page 11, lines 1-11).**

Furthermore, Figure S14 exhibits various Cu-Zn₅-ZrO₂ models; however, the computational approach employed to explore these structures for a comprehensive screening of structural possibilities is unspecified.

We compared all the potential energies of all the configurations in **Supplementary Fig. 14** according to **Supplementary Equations 1-2**. The most stable configuration comprising 5 incorporated Zn atoms and 5 oxygen vacancies was then used to study single Cu adsorption. Based on this optimum model Cu-Zn₅ZrO_x structures with 2 Cu atoms were built, and the same procedure was followed for 3 and 4 Cu atoms. **We have added the potential energies of all structures shown in Supplementary Fig. 14 to Supplementary Table 5, and included a snapshot of each model in Supplementary Fig. 20. The computational approach has also been clarified in the caption of Supplementary Table 5 (page 12, lines 5-7).**

Finally, the Supplementary material, indicates (page 3) that the formation of larger clusters is unfavorable. Yet, based on the energies depicted in Figure S14, even the incorporation of the first Cu atom into Zn₅-ZrO₂ structure appears to energetically unfavorable. Overall, clarity regarding these structural models and their outcomes is necessary for comprehensive understanding.

The adsorption of a single Cu atom on Zn₅ZrO_x (**Fig. 5**) is more favorable than on zinc-free ZrO₂ surfaces by 1.31 eV. Additionally, the adsorption of a second Cu atom is particularly favored by 0.50 eV, resulting in a CuZn_x motif akin to Cu₅Zn₈(110), and in line with experimental characterization of the Cu-ZnZrO_x catalyst. Moreover, models with 3 and 4 Cu atoms are less stable in terms of potential energies (**Supplementary Fig. 14**). Therefore, the Cu₂-Zn₅ZrO_x structure was selected as the representative model to assessing the catalytic properties of Cu-ZnZrO_x computationally. **We have further clarified this point in the supplementary information (page 6, lines 8-28 and page 7, lines 1-2).**

2. Figure 7 provides a summary of the adsorption energies of key experimentally identified intermediates. I suspect that the adsorption energies refer to the energetically most favorable energies although this is not said in the figure caption. It would be beneficial to include the corresponding adsorption structures for CO, HCOO and CH₃O structures at least in Supplementary material.

We assessed the adsorption energies, E_{ads} , of different conformations of key reactants, intermediates, and product species on several models to rationalize the catalytic properties of Cu-ZrO_x, ZnZrO_x, and Cu-ZnZrO_x. **Fig. 7** presents the most favorable E_{ads} on the most relevant models while detailed E_{ads} values and adsorption conformations for other models are shown in **Supplementary Fig. 19**. **We have clarified this point in the revised manuscript (page 14, lines 6-9) and in the caption of Fig. 7. We have also added a new figure to the supplementary information (Supplementary Fig. 26), which contains snapshots of the adsorption of key species on the most relevant models corresponding to the E_{ads} depicted in Fig. 7.**

There are notable concerns regarding these adsorption energies. Firstly, the computational approach lacks the consideration of VdW corrections, neglecting weak long-range interactions between the adsorbate and the surface. This omission can potentially introduce errors into calculated adsorption energies.

To evaluate the effect of weak long-range interactions between adsorbate and surface on key adsorption energies shown in **Fig. 7**, we have applied van der Waals corrections using Grimme's D3 approach (see S. Grimme et al., *J. Comput. Chem.* **2011**, 32, 1456). **Two new tables comparing the E_{ads} values obtained through different computational approaches were added to the Supplementary information (Supplementary Tables 8 and 9).** The overstabilization induced by van der Waals contributions are systematic. **We have included this discussion in the supplementary information**

(page 16, lines 4-8) and amended the Computational details section in the revised manuscript (page 26, lines 1-10).

Second, the cooperative effects present for adsorption on oxides are unsystematically considered in the present work, please see J. Phys. Chem. C 121, 18608 (2017). According to the computational details, the adsorption of HCOO and H₃CO is computed in the presence of surface H, while no surface-H is present for CO₂ and CO adsorption. This discrepancy considering surface-H presence or absence can lead to significant differences in adsorption energies of carbon-based species. Consequently, in this study, the computed adsorption energies might not directly comparable due to varying conditions.

In the original submission, we simulated the adsorption of HCOO and CH₃O in the presence of surface-H to avoid adsorption of fragments with unpaired electrons. This fulfills the criteria of E. Castelli *et al.*, (*J. Phys. Chem. C* **2017** 121, 18608) stating that due to cooperative effects, the adsorption of acceptor-donor pairs on semiconducting metal oxides is more favorable compared to unpaired fragments. When an acceptor-donor pair is adsorbed, electrons are transferred from the donor to the acceptor through the semiconductor surface without altering the electronic properties since the total number of electrons remains unchanged. In contrast, adsorption of an unpaired fragment on the semiconductor surface induces electron transfer from the adsorbate to the surface or vice versa, resulting in a modification of the semiconductor's electronic properties. Therefore, all our computed adsorption energies maintain the charge balance needed (**Table R1**).

Table R1. Number of valence electrons used to determine adsorption energies depicted in Fig. 7.

Model	Number of valence electrons					
	Surface	CO ₂	H ₂	CO	HCOO*+H*	CH ₃ O*+H*
m -ZrO ₂ (-111)	1536	1552	1538	1546	1554	1550
Cu(111)	704	720	706	714	722	718
ZnZrO _x	378	394	380	388	396	392
Cu ₂ -Zn ₅ ZrO _x	1528	1544	1530	1538	1546	1542

We have added the reference pointed by the Reviewer and amended the Computational details section to further highlight the need to evaluate the adsorption of HCOO and CH₃O in the presence of surface-H (page 26, lines 1-10).

It is important to note that the presence or absence of terminal OH groups formed during the reaction could also influence the acid-base balance and subsequent adsorption energies. These factors collectively warrant attention for a more comprehensive evaluation of the adsorption phenomena.

As suggested by the Reviewer, the presence of hydroxyl groups (OH) near the metal ensemble will reduce the number of basic centers to adsorb CO₂. However, their concentration will be controlled by the steady-state conditions. Hence, given the thermoneutral nature of water elimination and vacancy formation (−0.3 to 0.2 eV), a low accumulation of OH is expected. **We have now assessed the potential influence of OH terminal groups on the adsorption energy of CO₂, H₂, and CO at the Cu₂-Zn₅ZrO_x ensemble.** The adsorption energies are within 0.1 eV and, therefore, the effect of OH terminal groups would be relatively small. **We have clarified this point in the supplementary information (page 6, lines 23-27 and page 7, lines 1-2).**

Furthermore, the choice of excluding f correction from the calculations must be computationally justified.

We have computed the adsorption energies of key intermediates shown in Fig. 7 by including a $U_{\text{eff}} = 4.00$ eV for d -states in Zr as suggested (see S. T. Korhonen *et al.*, *J. Phys. Chem. C* **2012**, 116, 6636; R. Grau-Crespo *et al.*, *J. Phys. Chem. C*, **2016**, 120, 17604; H. Koga *et al.*, *Appl. Surf. Sci.* **2020**, 508, 145252; X. Tang *et al.*, *Surf. Sci.* **2022**, 716, 121976). The adsorption energies for CO_2 , H_2 , CO , HCOO , and CH_3O do not change significantly. **This discussion has been included in the supplementary information (page 16, lines 4-8) and the Computational details section has been amended in the revised manuscript (page 26, lines 1-10).** and supplementary information (Supplementary Tables 8 and 9).

3. Calculations have exclusively focused on thermodynamic aspects of methanol synthesis. I warmly recommend extending these computations to include the determination of activation energies, specifically encompassing H_2 dissociation as well as the hydrogenation of HCOO and H_3CO . This broader computational scope will not only enhance the validation of catalyst models, but also provide additional support for the experimental findings.

We extended our computational analysis to determine the full energy profiles for CO_2 hydrogenation to methanol and CO on the relevant models presented in Fig. 5 (Supplementary Fig. 29-30). Our focus on the formate pathway aligns with previous studies on ZnZrO_x (see J. Wang *et al.*, *Sci. Adv.* **2017**, 3, 170129; K. Lee *et al.*, *Appl. Catal. B Environ.* **2022**, 304, 120994; *Adv. Energy Mater.* **2023**, 13, 2204122) and other oxide-based catalysts (see *J. Catal.* **2018**, 361, 313; *Nat. Commun.* **2019**, 10, 3377; *Adv. Energy Mater.* **2022**, 12, 2103707). Operando DRIFTS measurements (Fig. 8) identified HCOO^* and CH_3O^* species on Cu-ZrO_x , ZnZrO_x , and Cu-ZnZrO_x systems, further supporting our investigation of methanol synthesis through the formate pathway. Finally, since gas-phase CO was also detected for Cu-ZrO_x and Cu-ZnZrO_x (Supplementary Fig. 28), **we have also evaluated methanol formation on the $\text{Cu}(111)$ surface via CO hydrogenation.**

Methanol formation is favored over CO production on $m\text{-ZrO}_2(-111)$, ZnZrO_x , and $\text{Cu}_2\text{-Zn}_5\text{ZrO}_x$ (Supplementary Fig. 29-30), in line with experiments (Fig. 3). In contrast, the energy profiles computed on $\text{Cu}(111)$ indicate that CO_2 hydrogenation to methanol competes with CO formation. Still, CO desorption is favored compatible with the lower S_{MeOH} of Cu-ZrO_x . The higher methanol productivity exhibited by Cu-ZnZrO_x compared to Cu-ZrO_x and ZnZrO_x catalysts can be rationalized by their distinct ability to activate CO_2 and H_2 (Fig. 7). H_2 activation is barrierless and CO_2 is effectively activated at a nearby site on $\text{Cu}_2\text{-Zn}_5\text{ZrO}_x$, thus avoiding long-range transport of species. Likewise, ZnZrO_x activates CO_2 and H_2 at spatially resolved sites, but H_2 activation has a small barrier ($E_a = 0.21$ eV) and the adsorption of both reactants is less favored than on $\text{Cu}_2\text{-Zn}_5\text{ZrO}_x$, thus explaining the lower methanol productivity of ZnZrO_x . For Cu-ZrO_x , H_2 activation energies are 0.70 and 0.58 eV for $m\text{-ZrO}_2(-111)$ and $\text{Cu}(111)$, respectively. However, the adsorption energy of H_2 is exothermic ($E_{\text{ads,H}_2} = -0.40$ eV) on $\text{Cu}(111)$, while it is endothermic ($E_{\text{ads,H}_2} = 0.34$ eV) over $m\text{-ZrO}_2(-111)$. These results indicate that CO_2 is activated on the ZrO_2 support while H_2 is split on the Cu nanoparticles. Hence, the required transport of active species and the poor ability to activate H_2 most likely explain the lower methanol productivity of Cu-ZrO_x . **We have incorporated these findings in the revised manuscript (page 17, lines 8-26, page 15, lines 2-12), including the computed energy profiles for the different systems (Supplementary Fig. 29-30).**

4. Minor issues: 1) When discussing heterolytic H_2 dissociation, Figure 9 should be cited not Figure 7.

We have cited Fig. 9 in the revised manuscript when discussing heterolytic H_2 dissociation (page 15, lines 5, 12) and thank the Reviewer for raising this point.

Reviewer #3

The manuscript reported a metal promotion in ZnZrO_x catalyst used for CO₂ hydrogenation to methanol. It was found that Cu had the best effective. The authors suggested that the low-nuclearity CuZn_x clusters on the ZrO₂ surface are responsible for the effect. The manuscript could be published after addressing the comments

We warmly thank the Reviewer for their positive remarks and constructive comments.

1. The promoted Cu concentration should be optimized in this work.

We appreciate the Reviewer's valuable suggestion. Optimizing the Cu content in the ternary Cu-ZnZrO_x systems is indeed valuable. However, our current focus is on fundamental aspects of metal promotion. Further optimization of the composition will be undertaken in future work. **We have justified this in the revised manuscript (page 18, lines 7-8).**

2. CuZnO_x catalyst should also be placed to the comparison object.

We thank the Reviewer for raising this excellent point. Accordingly, **we have synthesized a reference CuZnO_x catalyst by FSP and evaluated its performance**, which is comparable to Cu-ZrO_x but notably inferior to Cu-ZnZrO_x, further highlighting the superior catalytic properties of the ternary system. **We have included the new result (Supplementary Fig. X) and corresponding discussion in the supplementary information and revised manuscript, respectively (page 8, lines 12-15).**

3. Is there additional evidence to support or affirm the Cu structure in the Cu-ZnZrO_x catalyst. Specifically, single atom, di-atom, or cluster? By contrast, Pd seems to in the forms of particale?

In our original paper, we have already exhausted both microscopy and *operando* spectroscopy techniques available to characterize copper speciation on the Cu-ZnZrO_x catalyst. However, due to their inherent limitations, we can only confirm that copper predominantly associates with zinc, forming low-nuclearity clusters. Additionally, the 10 times lower copper content compared to zinc and the high methanol selectivity of Cu-ZnZrO_x suggest the presence of fewer copper atoms in these clusters. Guided by these experimental observations, **we have performed additional DFT calculations upon revision, exploring various sizes and 56 configurations of the CuZn clusters (Supplementary Fig. X and Tables Y and Z).** These new results indicate that models with 2 copper (possibly forming dimer species) and 5 zinc atoms are more likely thermodynamically. Therefore, the Cu₂-Zn₅ZrO_x structure was selected as the most representative model for evaluating the catalytic properties of Cu-ZnZrO_x catalysts from a computational standpoint. **We have elaborated on the discussion of the copper speciation in the revised manuscript (page 11, lines 23-26 and page 12, lines 1-6).** With respect to palladium, both microscopy and *quasi in situ* XAS experiments confirm that it forms nanoparticles (see **Fig. 2** and **Supplementary Fig. 10**).

4. It can be seen that Cu-ZnZrO_x catalyst exhibits less stability than ZnZrO_x catalyst?

This point is in line with **Comment #1** of **Reviewer #1**. ZnZrO_x reaches steady-state methanol productivity almost instantaneously, whereas Cu-ZnZrO_x requires *ca.* 50 h on stream (**Fig. 3a**). However, since catalytic performance remains unchanged for thereafter, we would not classify this phenomenon as lower stability; rather, it reflects the typical induction period needed for the catalyst to restructure and equilibrate until reaching stable performance. Therefore, it is reasonable to conclude that both systems are remarkably stable, but distinct induction times are necessary to attain a steady-state performance, with Cu-ZnZrO_x requiring a longer time compared to ZnZrO_x, likely due to the

more complex architecture of the former catalyst. **We have included this remark in the revised manuscript (page 8, lines 4-9).**

5. Weather the Cu promotion changed the pathway from CO₂ to methanol? HCOO pathway or CO* pathway?*

As highlighted in the original manuscript, *operando* DRIFTS experiments were conducted on Cu-ZrO_x, ZnZrO_x, and Cu-ZnZrO_x systems to gain insights into key intermediates formed during CO₂ hydrogenation. The three catalysts exhibited signals that can be assigned to HCOO* and CH₃O* species, indicating that methanol formation proceeds *via* the formate pathway. We have further corroborated these experimental findings by **computing the energy profiles for CO₂ and CO hydrogenation to methanol and CO formation *via* the reverse water-gas shift (RWGS) reaction (Supplementary Fig. X-Y) on the most relevant models presented in Fig. 5 (Cu(111), *m*-ZrO₂(-111), ZnZrO_x, and Cu₂-Zn₅ZrO_x).** As discussed in the response to **comment#3** of **Reviewer#2**, methanol formation is favored over CO on *m*-ZrO₂ (-111), ZnZrO_x and Cu₂-Zn₅ZrO_x. Additionally, CO desorption is favored compared to its further hydrogenation to CH₃OH, indicating that copper promotion does not change the preferred reaction pathway for CO₂ hydrogenation to methanol over Cu-ZnZrO_x. **We have further highlighted this point in the revised manuscript (page 17, lines 8-25).**

REVIEWERS' COMMENTS

Reviewer #1 (Remarks to the Author):

The authors have effectively addressed my questions and doubts, and as a result, I highly recommend this paper for publication in this journal.

Reviewer #3 (Remarks to the Author):

The authors mentioned the non-solid solution of ZnZrO structure prepared by flame method, and emphasized the effect of CuZn clusters. Thus, I also want to know the catalytic performances of a reference catalyst CuZn-Al₂O₃

In addition, there is a mistake in Supplementary Fig. 11, the second sample is Cu-ZnO_x, rather than Cu-ZrO_x, am I right?

Response to Reviewers for Manuscript NCOMMS-23-45274A

Code: Comments in blue | Replies in black | Actions in bold

Note: Indicated page, line, figure, or reference numbers refer to the revised manuscript and/or supplementary information with changes highlighted.

Reviewer #1

The authors have effectively addressed my questions and doubts, and as a result, I highly recommend this paper for publication in this journal.

We thank Reviewer #1 for their inputs to strengthen our manuscript in the previous revision and for supporting its publication.

Reviewer #3

The authors mentioned the non-solid solution of ZnZrO structure prepared by flame method, and emphasized the effect of CuZn clusters. Thus, i also want to know the catalytic performances of a reference catalyst CuZn-Al₂O₃.

As now further clarified in the manuscript (page 2, line 18), the commercial CuZn-Al₂O₃ reference has already been reported (Li, K. & Chen, J. G., *ACS Catal.* 9, 7840-7861, 2019) and is known to be unselective for CO₂ hydrogenation. Our manuscript demonstrates the importance the concept of copper promotion, leading to highly selective performance.

In additional, there is a mistake in Supplementary Fig. 11, the second sample is Cu-ZnOx, rather than Cu-ZrOx, am i right?

Thank you noticing this typo, **which we have corrected in the revised manuscript.**